# ZFP30 promotes adipogenesis through the KAP1-mediated activation of a retrotransposon-derived *Pparg2* enhancer

Wanze Chen[1,2,7], Petra C. Schwalie[1,2,7], Eugenia V. Pankevich[1,3], Carine Gubelmann[1,2], Sunil K. Raghav[1,4], Riccardo Dainese[1,2], Marco Cassano [5], Michael Imbeault[5], Suk Min Jang[5], Julie Russeil[1,2], Tenagne Delessa[6], Julien Duc[5], Didier Trono[5], Christian Wolfrum[6] & Bart Deplancke [1,2]

Krüppel-associated box zinc finger proteins (KZFPs) constitute the largest family of mammalian transcription factors, but most remain completely uncharacterized. While initially proposed to primarily repress transposable elements, recent reports have revealed that KFZPs contribute to a wide variety of other biological processes. Using murine and human in vitro and in vivo models, we demonstrate here that one poorly studied KZFP, ZFP30, promotes adipogenesis by directly targeting and activating a retrotransposon-derived *Pparg2* enhancer. Through mechanistic studies, we further show that ZFP30 recruits the co-regulator KRAB-associated protein 1 (KAP1), which, surprisingly, acts as a ZFP30 co-activator in this adipogenic context. Our findings provide an understanding of both adipogenic and KZFP-KAP1 complex-mediated gene regulation, showing that the KZFP-KAP1 axis can also function in a non-repressive manner.

[1] Laboratory of Systems Biology and Genetics, Institute of Bioengineering, School of Life Sciences, Ecole Polytechnique Fédérale de Lausanne (EPFL), CH-1015 Lausanne, Switzerland. [2] Swiss Institute of Bioinformatics (SIB), CH-1015 Lausanne, Switzerland. [3] Faculty of Bioengineering and Bioinformatics, Lomonosov Moscow State University, 119234 Moscow, Russian Federation. [4] Immunogenomics & Systems Biology group, Institute of Life Sciences, Bhubaneswar 751023 Odisha, India. [5] Laboratory of Virology and Genetics, Global Health Institute, School of Life Sciences, Ecole Polytechnique Fédérale de Lausanne (EPFL), CH-1015 Lausanne, Switzerland. [6] Institute of Food Nutrition and Health, Eidgenössische Technische Hochschule Zürich (ETHZ), CH-8603 Schwerzenbach, Switzerland. [7] These authors contributed equally: Wanze Chen, Petra C. Schwalie. Correspondence and requests for materials should be addressed to B.D. (email: bart.deplancke@epfl.ch)

KZFPs constitute the largest mammalian transcription factor (TF) family[1], with about 400 and 600 members (including pseudogenes) in human and mouse, respectively[2,3]. KZFPs are characterized by the presence of an N-terminal Krüppel-associated box (KRAB) domain and an array of the C-terminal C2H2 zinc finger domains[4]. The KRAB domain typically recruits the co-repressor KRAB-associated protein 1 (KAP1, also known as TRIM28, Tif1β), which serves as a scaffold protein for the further recruitment of co-repressors[5]. The C2H2 zinc finger domains determine a KZFP's DNA-binding specificity, with each zinc finger typically, but not always[6], recognizing three DNA bases[7]. The amino acids in positions 1, 2, 3, and 6 in the alpha helical region of each zinc finger directly interact with the DNA, thus acting as the most critical binding specificity determinants[8,9], referred to as *fingerprint*[10]. The KZFPs with similar fingerprints are likely to recognize similar DNA sequences. However, the exact DNA-binding specificity of most KZFPs is yet to be determined.

Despite their abundance, most KZFPs have not been functionally characterized[4]. KZFPs have been shown to play roles in some processes, such as genomic imprinting, cell differentiation, metabolic control, and sexual dimorphism[1]. However, since the KRAB-binding co-repressor KAP1 and several KZFPs were found to be essential for the repression of transposable elements (TEs) in early embryonic development[11–17] and also adult tissues[18,19], it has been proposed that the primary role of KZFPs is to control TE invasion.

We have recently uncovered one poorly studied KZFP, ZFP30, as a top hit in a large-scale TF overexpression screen aimed at identifying novel pro-adipogenic regulators[20]. This finding suggests that ZFP30 may be a thus far unrecognized component of the adipogenic regulatory network. The latter is one of the better characterized differentiation networks, as many years of research using especially in vitro cellular models such as 3T3-L1 cells[21] have greatly advanced our understanding of adipogenic gene regulation. In essence, the principal network components are the TFs C/EBPβ and C/EBPδ that activate the expression of the adipogenic master regulators C/EBPα and PPARγ[22,23]. The latter TF, which is present as two isoforms (PPARγ1 and PPARγ2) with only PPARγ2 being restricted to adipose tissue[24], is both necessary and sufficient for adipogenesis since it directly regulates the majority of adipocyte-specific genes[25]. This may explain why *Pparg* is itself regulated by a complex network of pro-adipogenic TFs[26–28].

It is clear, however, that our understanding of the adipogenic regulatory network is still far from complete, as also revealed by our recent genome-wide screen that identified several unexpected adipogenesis-regulating TFs[20]. In particular, the KZFP ZFP30 stood out as one of the top adipogenesis-enhancing candidates. Here, we demonstrate its critical role in adipogenesis using murine in vitro and in vivo models, as well as human stromal vascular fraction (SVF) cells. Interestingly, while ZFP30 targets retrotransposons and acts through KAP1, consistent with our canonical understanding of KZFP action, its role is to activate rather than to repress *Pparg2* expression and thus adipogenesis. We further show that this surprising activating role of the ZFP30–KAP1 complex is dependent on ZFP30-mediated recruitment to an ancient retrotransposon located upstream of the *Pparg2* promoter. Together, our results provide a functional characterization of the KZFP ZFP30, revealing its target landscape, DNA-binding specificity, detailed mode of action at the *Pparg2* locus in the context of adipogenesis, and evolutionary relation with specific retrotransposons.

## Results

**ZFP30 is a positive regulator of adipogenesis.** The KZFP ZFP30 ranked second among the endogenously expressed TFs enhancing 3T3-L1 fat cell differentiation[20]. We thus explored (1) whether ZFP30's role in adipogenesis is general and (2) what the underlying regulatory mechanisms are. To validate the screen data, we reduced *Zfp30* expression by lentivirus-mediated shRNA in 3T3-L1 cells and then induced adipocyte differentiation (see the Methods section). We observed striking differences in lipid accumulation (as assessed by Oil Red O—ORO—staining) (Fig. 1a; Supplementary Fig. 1A), which correlated with *Zfp30* expression levels (lipid accumulation versus relative *Zfp30* expression, Pearson's $r = 0.94$, $p = 0.017$, Supplementary Fig. 1A). Consistently, the expression level of the adipogenic marker genes *Pparg2*, *Adipoq*, and *Fabp4* were significantly lower ($p < 0.01$, $t$ test) in *Zfp30* knockdown (KD) cells compared with the control (Fig. 1b).

We next generated *Zfp30* knockout (KO) cells using CRISPR/Cas9[29,30]. However, we found that both KO and wild-type cells generally lost their differentiation capacity, potentially due to their extended culture upon expansion from single clones[31]. Given that the core white and brown adipogenic differentiation programs are largely shared[32], we isolated a clonal subline from an immortalized murine-derived brown pre-adipocyte cell line (IBA)[33]. We found that these subcloned IBA cells constitute a suitable model alternative to 3T3-L1 cells given that they (i) do not express the brown adipocyte marker gene *uncoupling protein 1* (*Ucp1*) when differentiated (Supplementary Fig. 1B), (ii) accumulate large amounts of lipids when induced with the adipogenic induction cocktail MDI (3-isobutyl-1-methylxanthine, dexamethasone, and insulin), and (iii) maintain a high differentiation potential even after long-term culture. Using CRISPR/Cas9, we successfully obtained four *Zfp30* mutant clones (Supplementary Fig. 1C). Based on remaining *Zfp30* mRNA expression levels (Supplementary Fig. 1D), these are expected to express largely truncated, loss-of-function proteins, which is why we annotate them as *Zfp30* KO cells. We also generated five WT clones as controls. *Zfp30* mRNA levels in the WT cells were maximal at confluence, sharply decreased upon MDI exposure, almost fully recovered at day 1, and finally stabilized at about half of the initial value after differentiation day 2 (Supplementary Fig. 1D). After adipogenic differentiation, four out of the five WT and none of the KO IBA clones accumulated large amount of lipids (Fig. 1c; Supplementary Fig. 1E). The distinct differentiation capacity in KO and WT was not driven by differences in cell growth or proliferating rate, since the cell numbers were comparable after differentiation (Supplementary Fig. 1F). Consistently, the expression of adipogenic marker genes was significantly lower in *Zfp30* KO compared with WT cells (Fig. 1d; Supplementary Fig. 1G).

To investigate ZFP30 function in vivo, we attempted to generate *Zfp30* KO mice using *Zfp30*-targeted ES cells obtained from the European Conditional Mouse Mutagenesis Program[34] (*Zfp30^tm3e(EUCOMM)Hmgu*, termed *Zfp30^tm3e*) (Supplementary Fig. 1H). The homozygous mice (*Zfp30^tm3e/tm3e*) were viable but, surprisingly, *Zfp30* was still expressed in most of the tissues, including subcutaneous and visceral fat (Supplementary Figure 1I). Sequencing of the whole targeting cassette showed that all essential elements were intact. This suggests that the targeting cassette lost its function in this particular genome context, through a combination of bypassing the transcriptional termination site and splicing out of the construct. We therefore resorted to a previously described in vivo adipogenesis model[35] to test the function of ZFP30 in vivo. Briefly, SVF cells from mouse fat tissue were transduced with *Zfp30* or control shRNA. They were then embedded into Matrigel and transplanted into the subcutaneous layer of the mouse neck. The Matrigel pads were analyzed after subjecting the mice to 6 weeks of high-fat diet. We observed a significant ($p = 0.001$, paired $t$ test) reduction of the ratio of fat containing cells to the total cells in *Zfp30* KDs (Fig. 1e, f).

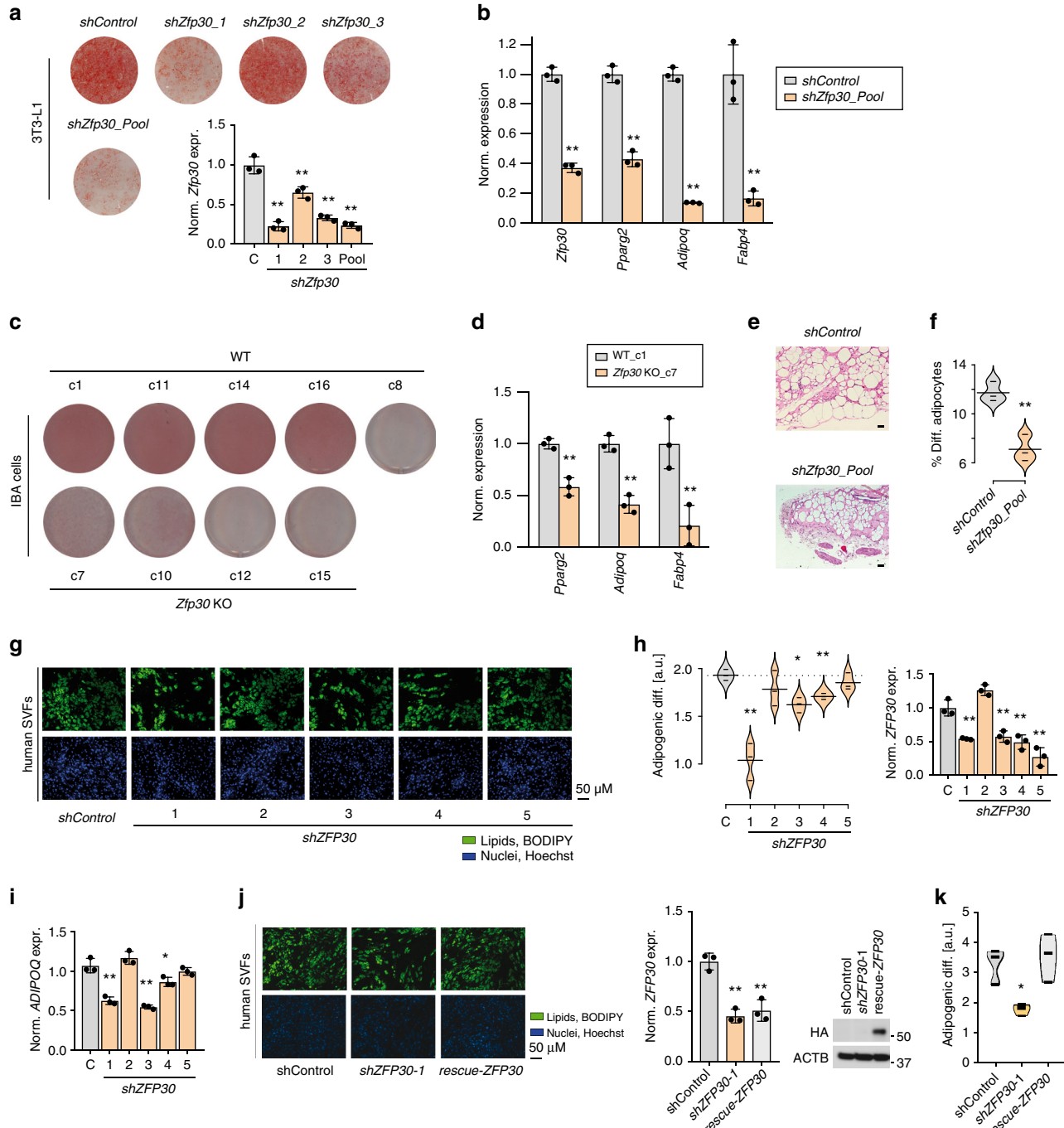

As ZFP30 is highly conserved between mouse and human (82% similarity, Supplementary Fig. 1J), with an almost identical fingerprint[10] (Supplementary Fig. 1K), we tested whether it also promotes adipogenesis in humans. We therefore isolated primary SVF cells from human lipoaspirate samples as previously described[36] and differentiated them with a hormonal cocktail, analogous to the mouse setup (see the Methods section). When ZFP30 levels were reduced by lentivirus-mediated shRNA, upon adipogenesis, we observed a significant ($p < 0.01$, $p < 0.05$, paired $t$ test) decrease in both lipid accumulation (based on BODIPY staining) (Fig. 1g, h) and adipogenic marker gene expression (Fig. 1i) for three out of the four shRNAs that successfully reduced ZFP30 expression. The observed difference was not driven by distinct proliferation rates, as overall cell numbers were comparable across differentiated samples (Supplementary Fig. 1L).

We then performed a rescue experiment by introducing an shRNA-resistant ZFP30 (code-optimized and without the 3′UTR targeted by shRNA-1), while endogenous ZFP30 was knocked down. As shown in Fig. 1j, k, the introduction of the shRNA-resistant ZFP30 restored the differentiation of human SVF cells to control levels, as assessed by lipid accumulation (BODIPY staining). In sum, we demonstrate that ZFP30 positively regulates adipogenesis in distinct murine in vitro and in vivo models as well as in ex vivo human cells.

**ZFP30 regulates multiple targets during adipogenesis**. To globally characterize ZFP30's target landscape, we performed transcriptomic analyses after Zfp30 reduction or absence, in 3T3-L1 and in IBA cells, respectively. We reasoned that shared

**Fig. 1** ZFP30 is a positive regulator of adipogenesis. **a** Effect of knocking down (KD) *Zfp30* or the negative control (shControl) on 3T3-L1 adipogenic differentiation as assessed by lipid accumulation (ORO staining). Corresponding *Zfp30* expression levels measured by qPCR are displayed below. **b** *Zfp30* and adipogenic marker gene expression upon *Zfp30* KD post adipogenic differentiation. $n = 3$ biologically independent experiments. **$p < 0.01$, $t$ test. **c** Effect of knocking out *Zfp30* on IBA adipogenic differentiation as assessed by ORO staining. Five wild-type (WT) and four KO clones are shown. **d** Adipogenic marker gene expression in control (c1) and *Zfp30* KO (c7) IBA cells post differentiation. $n = 3$ biologically independent experiments. **$p < 0.01$, $t$ test. **e**, **f** Adipocyte differentiation in stromal vascular fraction (SVF) transplants of three distinct mice fed a high-fat diet for 6 weeks (Methods). **e** Fat pad sections from representative samples of *Zfp30* KD and control SVF transplants stained with Haematoxylin (blue) and Eosin (pink). Scale bar: 100 μM. **f** Fat cell content (percent differentiated adipocytes) of the transplanted SVF cells containing *Zfp30* KD and control constructs; $n = 3$ biologically independent experiments. **$p < 0.01$, paired $t$ test. **g** Effect of *ZFP30* KD and control on primary SVF cells from human lipoaspirate samples as assessed by lipid accumulation (BODIPY, green; Hoechst nuclear staining, blue), scale bar 50 μm. Representative images of three independent experiments are shown. **h** The fraction of differentiated cells per each shRNA sample shown in (**g**) (left, **$p < 0.01$, *$p < 0.05$, paired $t$ test) and corresponding *Zfp30* expression levels (right) as assessed by qPCR. **i** *ADIPOQ* expression per each shRNA sample shown in (**g**). $n = 3$ biologically independent experiments. **$p < 0.01$, *$p < 0.05$, $t$ test. **j** Rescue of shZFP30 by introducing shRNA-resistant and code-optimized *ZFP30*. Experimental settings are similar to (**g**). Left: representative images are shown. Middle: the endogenous *ZFP30* mRNA level is shown by qPCR. Right: the exogenous HA-ZFP30 protein is shown by western blotting. **k** The fraction of differentiated cells for each sample shown in (**j**). $n = 3$ biologically independent experiments. *$p < 0.05$, $t$ test. All panels: error bars—SD. Source data are provided as a Source Data file

differences are most likely to be relevant for understanding the direct action of *Zfp30* on fat cell differentiation, removing unrelated cell type-specific effects. Since we already observed a significant ($p < 0.01$, $t$ test) reduction in adipogenic marker gene expression in *Zfp30* KD samples after 2 days of adipogenic induction (Supplementary Fig. 2A), we included day 0 and day 2 measurements in both 3T3-L1 and IBA cells, and two additional time points (2 h and day 4) in IBA cells. Importantly, we found that the core adipogenic gene program was highly upregulated in day 2 or 4 (2/4) compared with day 0 samples in both cell models (Supplementary Fig. 2B,C and Supplementary Data 1). Specifically, *Pparg*, *Plin4*, *Lipe*, and *Cd36* were among the top 30 induced genes upon differentiation (Supplementary Fig. 2B), and overall, the upregulated genes enriched for GO terms related to adipogenic functions (Supplementary Fig. 2C and Supplementary Data 2). This validates our earlier finding that both employed cell models are appropriate for studying the core adipogenic transcriptional program.

We quantified significant differences in terms of gene expression between WT and *Zfp30* KD/KO samples, at each individual time point (i.e., day 0, day 0, 2 h, day 2, and day 4) (DESeq2, FDR <= 0.1 and |FC| >= 2, Methods). We found that several hundreds of genes were deregulated when *Zfp30* levels were altered (Fig. 2a–f). In general, and most strikingly in KO samples, more differences were observed upon adipogenic differentiation, with over 600 genes being up- or downregulated, respectively, at days 2/4. Overall, we found similar proportions of up- and downregulated genes, with 150 genes consistently up- or downregulated in both KO and KD samples, suggesting a context-specific action of ZFP30 (Fig. 2a, d). Downregulated genes enriched for fat cell-related functionality, including GO terms such as brown fat cell differentiation, triglyceride biosynthetic process, and triglyceride homeostasis (Fig. 2b; Supplementary Data 2). The adipogenic marker genes *Adipoq*, *Cidec*, *Cebpa*, *Fabp4*, *Lipe*, *Retn*, and *Fmo1*, were among the top 30 genes most reduced in both KD and KO samples (Fig. 2c). In contrast, upregulated genes were not previously connected to adipogenesis, enriching for GO terms related to inflammatory, immune, and external stimuli response as well as ossification (Fig. 2e, f; Supplementary Data 2).

To distinguish between direct and indirect effects of ZFP30, we further set out to determine its genome-wide DNA-binding landscape using ChIP-seq. As neither the commercial ZFP30 antibodies nor four batches of customized ZFP30 antibodies recognized it specifically, as tested by immunoprecipitation, we performed ChIP-seq in 3T3-L1 cells expressing HA-tagged ZFP30 in a tetracycline-inducible manner, as also previously employed

for ZEB1[20]. The mRNA expression of exogenous *Zfp30* was induced to a similar level as the endogenous *Zfp30* by adjusting the amount of Doxycycline, to avoid potential artefacts due to protein overexpression. Our replicate HA-ZFP30 ChIP-seq experiments revealed limited binding (< 100 binding events, referred to as peaks) at day 0, but up to 400 peaks at day 2 of adipogenesis (Fig. 2g; Supplementary Fig. 2D) (Methods). Replicate experiments showed high ChIP enrichment correlations compared with IgG controls, with a clear separation of ChIP-ed factor first and days (0 vs. 2) second (Supplementary Fig. 2E and Supplementary Data 3).

To further confirm the specificity of our ZFP30 ChIP-seq assays, we reconstituted the *Zfp30* KO IBA cells with wild-type ZFP30 as well as with a mutant lacking the DNA-binding zinc finger domains (ZFP30-ΔZF) (Supplementary Fig. 2F). We observed two bands in the ZFP30-ΔZF mutant sample, indicating the potential presence of a post translational modification. We then performed ChIP assays on these cells using an anti-HA antibody and measured the binding enrichment at four distinct ZFP30-bound regions by qPCR. As shown in Supplementary Fig. 2G, the ZFP30-ΔZF mutant failed to bind to all of the four tested loci (the enrichment was indistinguishable from the negative control), experimentally validating both the reproducibility and specificity of the ZFP30 ChIP-seq data.

We found that the majority of regions targeted by ZFP30 were significantly more bound at day 2 compared with day 0 (Fig. 2h; Supplementary Fig. 2H and Supplementary Data 4), (FDR <= 0.15 and |FC| >= 2, Methods). Thus, while ZFP30 binding seems independent of terminal adipogenesis at some genomic locations, such as the *Oas1* gene cluster, many other genes such as *Glis3* and *Myof* are only bound by ZFP30 upon adipogenic induction (Fig. 2i; Supplementary Fig. 2I). This observation suggests that the overall chromatin structure of day 2 specifically bound regions may be more dynamic than that of other bound regions, as also recently described elsewhere[37]. Indeed, we found that day 2-specific ZFP30-bound locations were increasingly DNase I hypersensitive[38] across early adipogenesis (day 0 to 2 h, 4 h, and day 2), and also showed an increase in the enhancer-specific H3K4me1 histone modification[39] (Fig. 2j). These patterns were much less pertinent at invariant ZFP30-bound locations (Supplementary Fig. 2J).

To assess how many genes may be directly bound and regulated by ZFP30, we combined our ChIP-seq and RNA-seq datasets (Supplementary Fig. 2K). Overall, genes responding to *Zfp30* expression reduction were about two times more likely to be bound compared with all expressed genes, and genes repressed by ZFP30 at adipogenic day 2 were even more enriched for ZFP30 binding (Supplementary Fig. 2K).

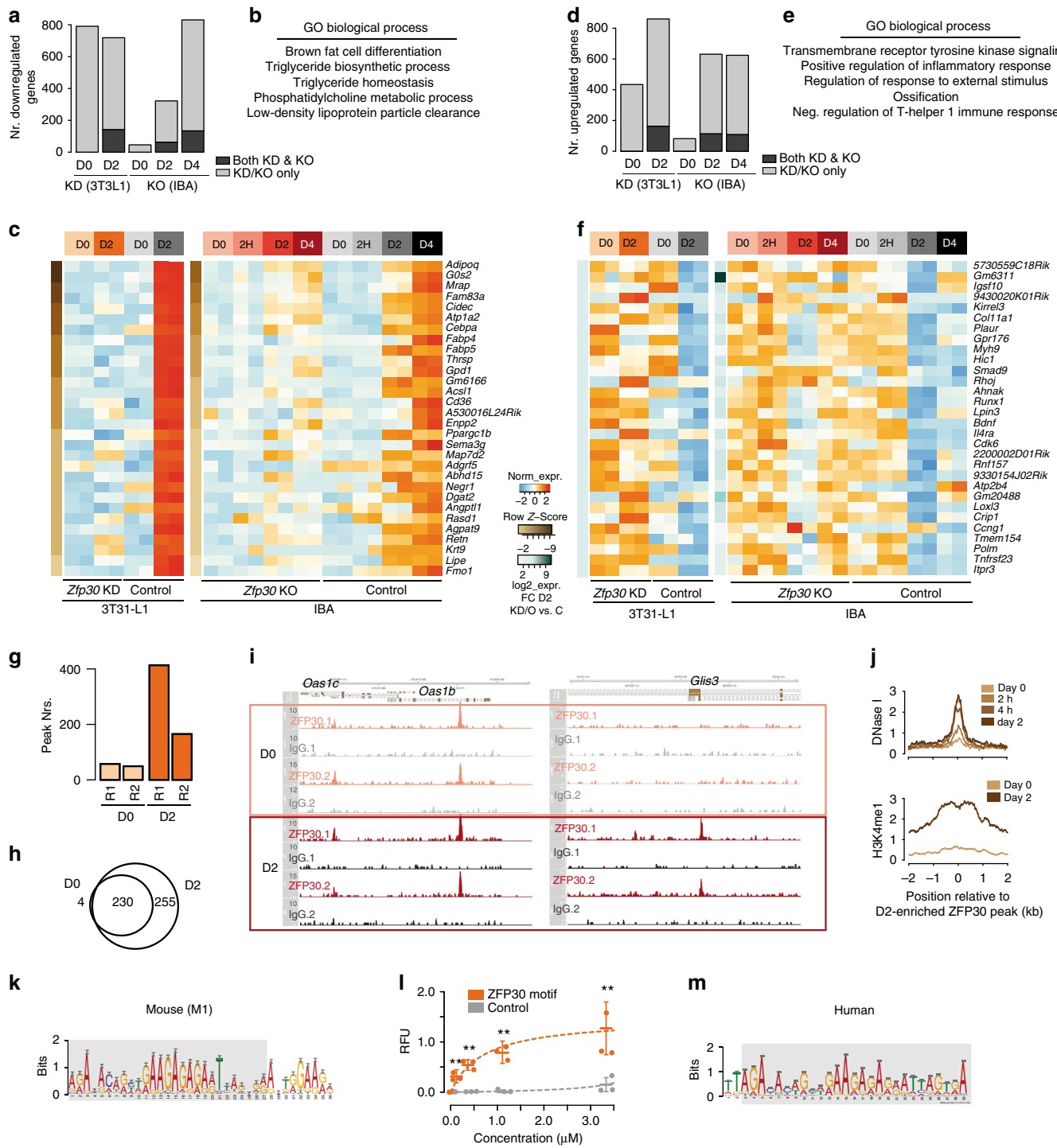

**Fig. 2** ZFP30 regulates multiple targets during adipogenesis. **a, d** Number of genes significantly (FDR 0.1, |FC| 2) down- (**a**) or up- (**d**) regulated upon *Zfp30* KD in 3T3-L1 cells at days 0 and 2 upon adipogenesis (left) as well as *Zfp30* KO in IBA cells at days 0, days 2, and 4 upon adipogenesis (right) (light gray). The number of commonly deregulated genes is depicted in dark gray. **b,e** Gene Ontology (GO) biological processes enriched among the genes commonly down- (**b**) and up- (**e**) regulated upon *Zfp30* KD and KO at days 2 or 4. **c, f** Heatmap displaying the top 30 genes commonly down- (**c**) and up- (**f**) regulated upon *Zfp30* KD and KO at days 2 or 4, ordered based on FCs in day 2 KD samples. **g** Number of ZFP30-HA ChIP-seq peaks in 3T3-L1 cells at days 0 and 2. Two biological replicates (R1 and R2) were performed. **h** The overlap between ZFP30-HA ChIP-seq peaks at days 0 and 2. **i** Examples of ZFP30-HA DNA binding at the *Oas1* gene cluster (adipogenic-invariant binding) and *Glis3* locus (adipogenesis-induced binding). **j** DNase I hypersensitivity (upper panel) and H3K4me1 (lower panel) signal upon adipogenic induction in a 4 kb window centered around the point of maximal ZFP30 binding, at locations bound by ZFP30 at day 2 only. **k** Top de novo motif (M1) discovered in DNA sequences bound by ZFP30 in mouse 3T3-L1 cells. **l** In vitro ZFP30 binding to DNA oligos bearing the consensus M1 motif as well as a control sequence, as assessed using MITOMI. The quantities of bound DNA are expressed in relative fluorescent units (RFU), normalized by protein amount. $n = 3$ biologically independent experiments. **$p < 0.01$, $t$ test. **m** Top de novo motif discovered in DNA sequences bound by ZFP30 in human HEK239T cells. All panels: error bars—SD. Source data are provided as a Source Data file

We next characterized the sequence specificity of the ZFP30-DNA interaction by performing motif discovery on genomic locations bound by ZFP30 in at least two samples across time points (Methods). The top two highly enriched motifs (termed M1 and M2, $E = 10^{-127}$ and $E = 10^{-52}$, AUC > 0.74, Fig. 2k; Supplementary Fig. 2L–O, and Methods) have not been previously described. As suggested by their similarity and the fact that their genomic matches are highly overlapping, M2 appears to be a more permissive variant of M1, as also suggested by its lower information content and the higher fraction of ZFP30 peaks that contain a significant M2 match (Supplementary Fig. 2O). To test if ZFP30 targets M1 directly, we performed a modified version of a mechanically induced trapping of molecular interactions (MITOMI) experiment[40,41]. We used the M1 consensus sequence as well as a control oligonucleotide across four distinct concentrations. We found that in vitro, the ZFP30 zinc finger domains significantly ($p < 0.01$, $t$ test) bind the M1 consensus motif, supporting the notion that ZFP30 binds M1-containing sites directly in vivo (Fig. 2l).

Given ZFP30's high human–mouse conservation (Supplementary Fig. 1J, K), one would expect it to show a similar DNA-binding specificity in humans. To test this, we resorted to the easy-to-use human HEK293T cells, considering that TF-binding motifs are in any case rarely tissue specific. Specifically, we applied ZFP30-HA ChIP-exo, since it tends to allow a greater resolution in identifying TF-binding sites, which benefits overall motif discovery (Methods). We then performed motif discovery on the 2582 detected bound genomic regions, similar to the above. We observed relatively broad peaks, which may reflect a suboptimal digestion by exonuclease (Supplementary Fig. 2P). However, both motif and ChIP-seq reads showed a strong central enrichment at these bound regions, suggesting the high specificity of our data (Supplementary Fig. 2P, Q). Indeed, the top enriched motif (Fig. 2m, $E = 10^{-415}$, hit rate = 64%, Methods) was a slightly shorter version of the mouse M1, which was, as expected, highly predictive of ZFP30 binding in mouse (AUC = 0.72, Methods) (Supplementary Fig. 2N).

Finally, we explored which TFs may collaborate in terms of DNA binding with ZFP30. To address this, we first performed an additional motif discovery analysis using HOMER findMotifs[42], which revealed significant enrichment of a range of adipogenesis-associated motifs in ZFP30-bound regions, including NF1, FOSL2, ATF3/4, and C/EBPbeta[20,23,27,32] (Supplementary Data 5). Second, we tested co-binding of ZFP30 with other TFs previously probed in 3T3-L1 cells[20,38,39,43,44]. We found that over one-fourth of ZFP30 peaks co-localized with adipogenic factors, such as C/EBPalpha, C/EBPbeta, ZEB1, and RXR, indicating potential co-regulation with these TFs (Supplementary Figure 2R).

**ZFP30 directly enhances *Pparg2* expression in adipogenesis.** One of the key functions of KZFPs is to target and modulate the transcriptional influence of transposable elements[4]. We therefore asked if ZFP30 also shows a similar behavior. Indeed, we found that both in mouse and human, ZFP30 binding is enriched in specific subsets of LINE1 elements, in particular L1MEg and L1MEd (Fig. 3a; Methods and Supplementary Data 6). Interestingly, L1MEd are among the oldest members of the L1 family, with an estimated age of ~150 million years[45]. It is thus conceivable that ZFP30, with its estimated age of ~105–150 million years[46] may have co-evolved with these L1 repeats. We then examined the TEs regulated by ZFP30 by analyzing the RNA-seq data from 3T3-L1 cells. While we found that there are ~600 TEs differentially expressed in *shZFP30* samples in both day 0 and day 2 (Supplementary Fig. 3A), none of them is bound by ZFP30. These findings suggest that ZFP30 does not act as a direct repressor of these still active TEs. This is further supported by the short length of ZFP30-bound L1s, suggestive of their decay (Supplementary Fig. 3B).

To gain better insight into ZFP30's role at its retrotransposon targets, we inspected expression patterns of genes proximal/overlapping L1-associated ZFP30 peaks (Fig. 3b). Similar to the general results reported above, we found that *Zfp30* KO/KD resulted in both up- and downregulation of nearby targets, suggesting context-specific effects. For instance, the genes in the *Oas* cluster, known for their role in immunity to viral infections[47], contained a bound L1MEd element and were upregulated in response to *Zfp30* KD. In contrast, the master adipogenic regulator *Pparg* contained an L1MC5a-associated ZFP30 peak and was significantly downregulated in the absence of ZFP30. Thus, suppression of *Zfp30* expression does not only result in upregulation but also in downregulation of proximal target genes. Given the old age of its target retrotransposons, it is likely that over time, at the mentioned locations, ZFP30 was eventually co-opted into the local gene regulatory network as either activator or repressor, after its transposable element-repressive role was no longer required due to the mutation-driven inactivation and divergence of the L1 elements.

To explore this hypothesis in greater molecular detail, we further investigated the ZFP30-targeted and regulated genes (Supplementary Fig. 2K). We found that while there were 79 genes bound and up- or downregulated in KD or KO samples (Supplementary Fig. 3C), which represents a significant enrichment ($p = 0.003$) compared to a randomized control (1000 repeats), 12 of them showed a consistent response across 3T3-L1 and IBA cells (Fig. 3c). While more genes bound and regulated by ZFP30 in both KO and KD samples were upregulated when *Zfp30* levels were reduced, we found that three known adipogenesis-associated genes were likely to be directly activated by ZFP30: *Cidec*, *Angptl4*, and most notably, *Pparg* (Fig. 3c).

To validate that ZFP30 regulates these genes directly, we performed luciferase reporter assays. We successfully cloned the bound DNA fragments (~700 bp) of *Plscr2*, *Tec*, *Col28a1*, *Myof*, *Ahnak*, *Pparg*, *Angptl4*, and *Cidec* into the pGL3-promoter vector. While the *Myof* fragment is derived from the promotor region, others are either intronic or distal intergenic regions. Reporter activities were measured in *Zfp30* KD and control 3T3-L1 cells, both in undifferentiated and differentiated (day 2) conditions using the empty pGL3-promoter vector as a negative/baseline control (Fig. 3d). Half of the reporters were consistently regulated irrespective of differentiation status, and the direction of regulation corresponded to the direction predicted based on the RNA-seq data. Specifically, *Pparg* and *Angptl4* reporter activity was lower upon *Zfp30* KD, whereas that of *Tec* and *Myof* was higher. Three other reporters (*Plscr2*, *Col28a1*, and *Ahnak*) showed differentiation-specific activity, while only *Cidec* showed no response in our assays. Together with the RNA-seq and ChIP-seq analyses, these results suggest that, while ZFP30 may indeed act most often in a repressive manner as expected from a KZFP, there are several genomic locations at which it activates transcription. Importantly, the latter includes the master adipogenic regulator, *Pparg*.

To validate further ZFP30's role in enhancing *Pparg* expression, we aimed to test its activity in a setting that maintains the regulatory element–promoter interaction[48]. ZFP30 binds a genomic site that is highly conserved across ~100 million years of evolution (human to armadillo), which was derived from a L1MC5a element, located about 9 kb upstream of the *Pparg2* transcription start site (Fig. 3e). In mouse, the larger region was disrupted by a more recent (rodent-specific) B3A element insertion, that occurred downstream of the ZFP30 motif, which itself remained intact. To validate that ZFP30 targets this specific motif instance, we again performed MITOMI experiments, this time using the exact genomic sequence at the *Pparg2* locus rather than the consensus motif, as well as a negative control. Our experiments revealed that this sequence is significantly ($p < 0.01$, $t$

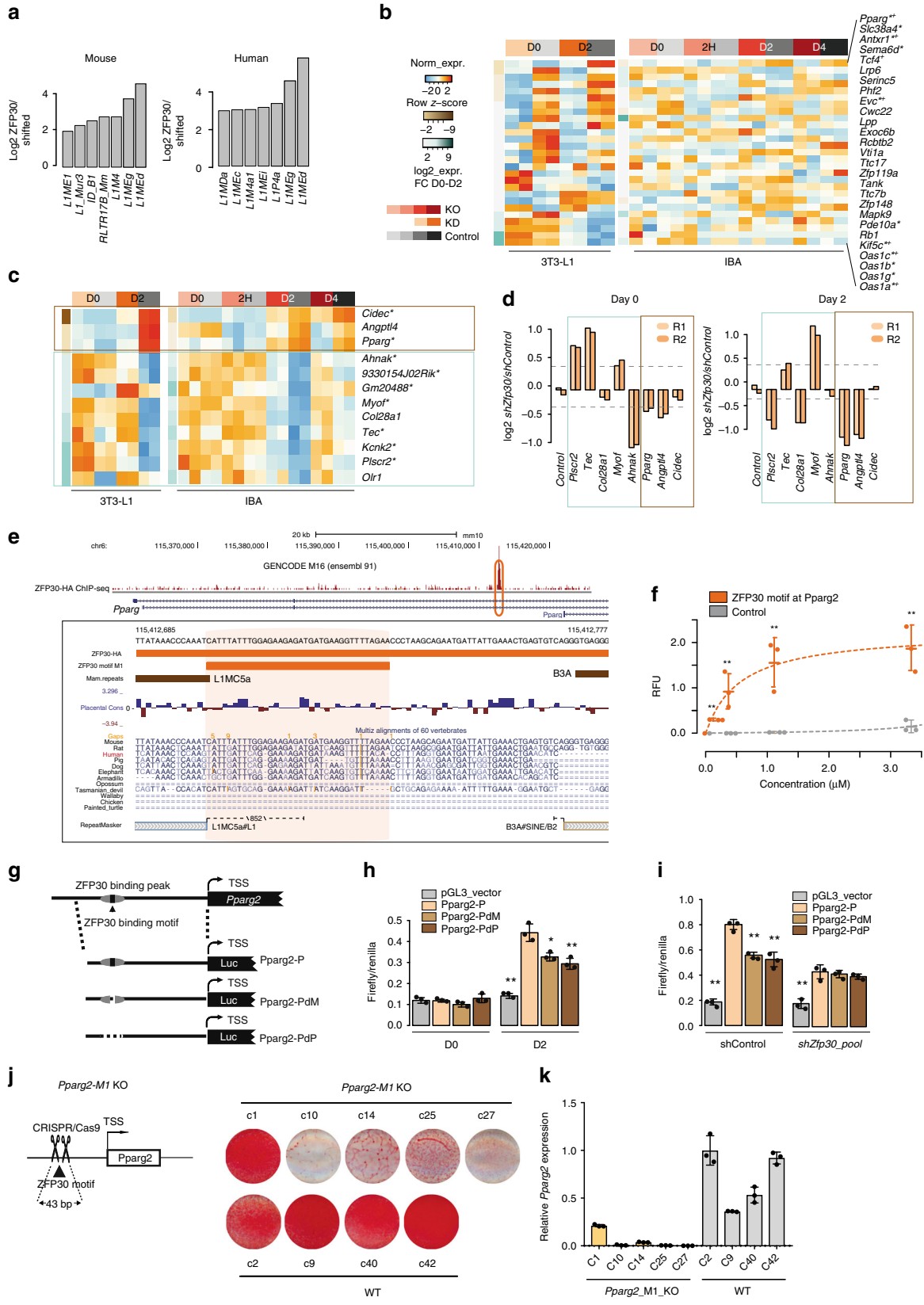

test) and directly bound by ZFP30's zinc finger domains in vitro (Fig. 3f). We then tested ZFP30's potential impact on local L1MC5a expression. Given the heavily decayed nature of this TE, we reasoned that the transcript derived from this TE would not be detected by our polyA-enriched RNA-seq protocol

(Supplementary Fig. 3A), even if expressed. This is why we performed RT-qPCR of this locus using a random hexamer for reverse transcription, revealing that the TE-derived transcript is indeed expressed, while being largely reduced in *Zfp30* KO cells (Supplementary Fig. 3D). These data indicate that ZFP30

**Fig. 3** ZFP30 directly enhances *Pparg2* expression during adipogenesis. **a** Enrichment of specific transposable element (TE) types among regions bound by ZFP30 in mouse 3T3-L1 (left) and human HEK293T (right) cells (Methods). **b** Heatmap displaying all genes showing proximal (< = 5 kb) TE-associated ZFP30 binding. *Significantly DE in *Zfp30* KD, + significantly DE in *Zfp30* KO at day 2 or 4. **c** Heatmap displaying all genes showing proximal ZFP30 binding and significant DE in both *Zfp30* KD and KO samples. *Day 2-specific ZFP30 binding. Same legend as in (**b**). **d** Luciferase assay showing the difference in regulatory activity upon *Zfp30* KD in 3T3-L1 (log2 shZfp30/shControl) for distinct sequences targeted by ZFP30. R1, R2—replicates 1 and 2. **e** Detailed visualization of the ZFP30-bound region upstream of the *Pparg2* promoter, including the ZFP30-HA ChIP-seq read pileup, motif, MITOMI-tested (in vitro) bound sequence (orange), vertebrate conservation and multiple alignment, as well as TEs (mammalian repeats Mam.repeats, top, brown, and UCSC RepeatMasker track, bottom, gray). **f** In vitro ZFP30 binding to the *Pparg2* motif match (dark orange in E) as well as to a control oligonucleotide as assessed by MITOMI. $n = 3$ biologically independent experiments. **$p < 0.01$, $t$ test. Note that the control curve is shared with Fig. 2l, as the binding assays were performed on the same MITOMI chips. **g** Schematic representation of the constructs employed for the luciferase assays displayed in (**h**, **i**). Pparg2-P: full construct; Pparg2-PdM: ZFP30 motif deletion; Pparg2-PdP: ZFP30 peak deletion. **h** Transcriptional activity as measured by luciferase assays (Firefly/ Renilla) for three distinct constructs shown in (**g**) as well as the control vector at days 0 and 2 of adipogenic differentiation (**h**), or at day 2 upon *Zfp30* KD (**i**). The values of all samples were compared with that of Pparg2-P in each group. $n = 3$ biologically independent experiments. **$p < 0.01$, *$p < 0.05$, $t$ test. **j** Schematic of *Pparg2*-M1 KO cells. **k** Left: effect of knocking out the ZFP30 DNA-binding motif at the *Pparg2* locus on IBA adipogenic differentiation as assessed by lipid accumulation (ORO staining). Right: corresponding *Pparg2* expression levels after differentiation were measured by qPCR. All panels: error bars—SD. Source data are provided as a Source Data file

directly binds to and activates the expression of this decayed L1MC5a TE.

Next, we cloned a ~9.5 -kb DNA fragment upstream of the *Pparg2* transcription start site (TSS) into a pGL30-basic luciferase vector (termed Pparg2-P), containing the promoter and the described ZFP30-bound region (peak) and motif. We also generated mutants lacking the ZFP30 peak (Pparg2-PdP) and ZFP30 motif (Pparg2-PdM), respectively (Fig. 3g). We found that the activity of the full-length reporter was induced when differentiated, but that of the Pparg2-PdP and Pparg2-PdM reporters was significantly ($p = 0.01$ and $p = 0.03$, respectively, $t$ test) reduced (Fig. 3h). Furthermore, we found that ectopic expression of ZFP30 tended to increase the activity of the Pparg2-P, but not the Pparg2-PdM reporter (Supplementary Fig. 3E). The inverse experiment was even more compelling, since the difference between the Pparg2-P and the Pparg2-PdM/Pparg2-PdP reporter activities disappeared when *Zfp30* mRNA levels were reduced by KD (Fig. 3i). Finally, to directly address whether this reflects the function of ZFP30 in its local endogenous genomic context, we used CRISPR/Cas9[29,30] to delete 43 bp of the Pparg genomic locus, including the ZFP30-binding motif and a few surrounding nucleotides, which, based on motif scanning, do not contain other relevant TF-binding sites (Fig. 3j; Supplementary Fig. 3F, Methods). As shown in Fig. 3k and Supplementary Fig. 3G, KO of the motif reduced lipid accumulation in four out of five clones of IBA cells, compared with the four wild-type clones. Consistently, *Pparg2*, *Pparg1* and *Adipoq* expression levels were also significantly reduced in these cells (Fig. 3k and Supplementary Fig. 3G). Collectively, these results strongly suggest that ZFP30 enhances *Pparg* gene expression by directly targeting a retrotransposon-derived enhancer element.

**KAP1 interacts with ZFP30 and promotes adipogenesis**. We set out to investigate how ZFP30 regulates its targets genes, with a particular focus on *Pparg*, given ZFP30's unexpected role in activating the expression of this adipogenic master regulator. As a KZFP (Supplementary Fig. 1C), ZFP30 is likely to interact with KAP1[5,49]. Indeed, we were able to co-immunoprecipitate ZFP30 with KAP1 when both were overexpressed in HEK293T cells (Fig. 4a). To further clarify whether this interaction also occurs in an adipogenic environment, we tested it in 3T3-L1 cells. As mentioned before, a ZFP30 antibody was not available; therefore, we used HA-ZFP30-expressing 3T3-L1 cells, as described previously for our ChIP-seq assays. Cells expressing the same empty vector and PPARG were included as controls. Endogenous KAP1 was pulled down together with HA-ZFP30,

but not with HA-PPARG, validating the interaction between the two proteins in an adipogenic setting (Figure 4b). The KRAB domain is well known to mediate the interaction of KZFPs with KAP1. Indeed, we found that the ZFP30 mutant containing a KRAB domain deletion (ΔKRAB) fails to interact with KAP1 (Fig. 4c). Interestingly, the mutant (ZFP30-In24aa) with a 24 amino acid insertion in the KRAB domain derived from the second allele of the *Zfp30* KO clone c15 (Supplementary Fig. 1C) also lost its KAP1-interacting capacity (Fig. 4c). These results experimentally confirm that ZFP30's KRAB domain is essential for its interaction with KAP1, in line with expectations based on the function of other KZFPs.

Given ZFP30's role as a pro-adipogenic regulator and the finding that the *Zfp30* KO clone c15, which expresses a KAP1-interaction-deficient *Zfp30* mutant, also shows low differentiation efficiency, we hypothesized that KAP1 has a similar adipogenic function to ZFP30. We thus tested this hypothesis by knocking down *Kap1* in 3T3-L1 cells. However, the KD led to dramatic cell death (Supplementary Fig. 4), consistent with previous reports in distinct cellular environments[50,51]. In our second model, IBA cells, we did not observe any significant cell death upon *Kap1* KD, supporting the notion that the effect of *Kap1* depletion is cell type-specific. We found that the differentiation of IBA cells was attenuated upon *Kap1* reduction, while reconstitution of *Kap1* expression with an shRNA-resistant KAP1 (Methods) restored the differentiation, as reflected by both lipid accumulation and adipogenic marker expression (Fig. 4d). We also generated primary mouse embryonic fibroblasts (MEF) cells from *Kap1*[loxp/loxp] mice. *Kap1* KO and control cells were generated from these cells by introducing the Cre recombinase and an empty vector, respectively. Although no significant lipid accumulation was observed in either of these cells upon an adipogenic stimulus, probably due to the variable differentiation capacity of MEF cells, consistent with previous reports[26], we detected a significant reduction of adipogenic marker gene expression in *Kap1* KO cells compared with controls (Fig. 4e). Together, these results demonstrate that ZFP30 interacts with KAP1, and KAP1 itself is required for adipogenesis across distinct cell types.

**ZFP30 activates *Pparg2* expression through KAP1**. To further characterize its role in adipogenesis and as a ZFP30 partner, we performed KAP1 ChIP-seq in undifferentiated (day 0) and dif-ferentiated (day 2) 3T3-L1 cells, as described above for HA-ZFP30. We first confirmed the high enrichment obtained with the employed KAP1 antibody (Supplementary Fig. 5A). Replicate experiments revealed ~300–500 KAP1-enriched regions genome-wide (Fig. 5a). Correlation analysis between the ChIP-seq enrichments showed an overall high correlation between the

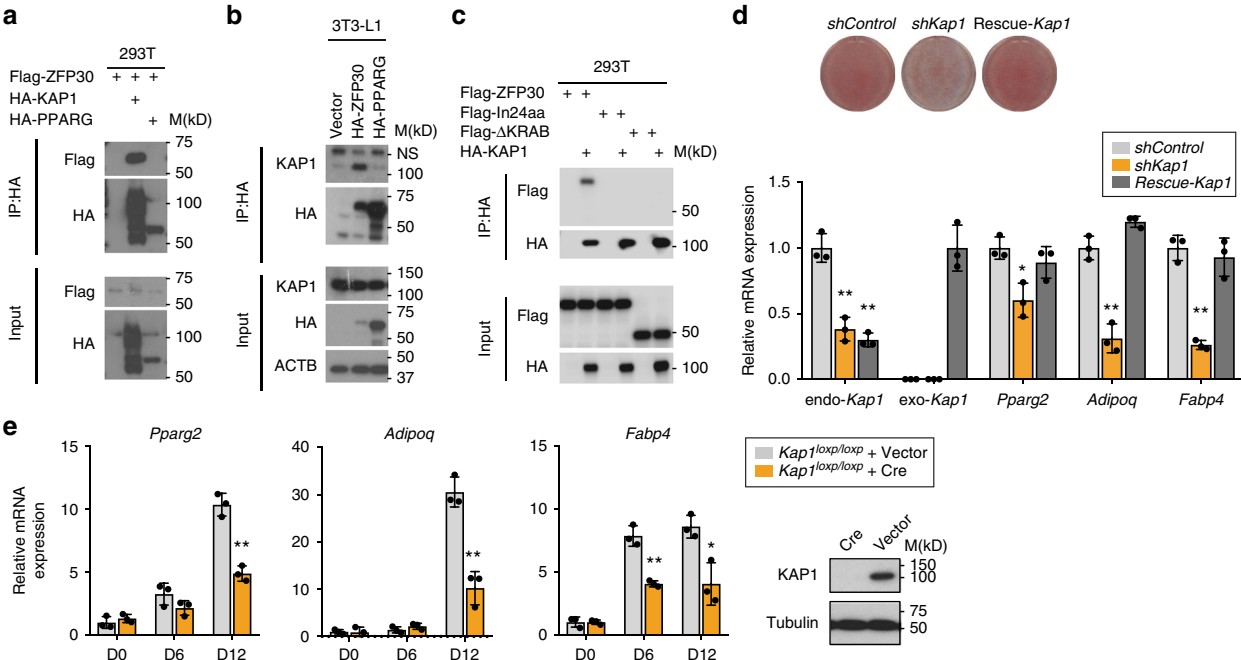

**Fig. 4** KAP1 interacts with ZFP30 and promotes adipogenesis. **a** Immunoprecipitation in HEK293T cells showing that ZFP30 interacts with KAP1. HEK293T cells transfected with the indicated constructs were subjected to immunoprecipitation with the HA antibody and subsequently probed with the listed antibodies. **b** Immunoprecipitation in 3T3-L1 cells showing that ZFP30 interacts with endogenous KAP1 in adipogenic conditions. 3T3-L1 expressing the indicated HA-tagged proteins were used for immunoprecipitating HA and subsequent western blotting with the listed antibodies. NS, non-specific band. **c** Immunoprecipitation in HEK293T cells showing that the KRAB domain of ZFP30 is essential for interaction with KAP1. ΔKRAB is a KRAB domain deletion mutant and In24aa is a mutant with a 24 amino acid insertion in the KRAB domain derived from the *Zfp30* c15 allele 2. **d** Effect of knocking down *Kap1* and of rescuing *Kap1* expression on IBA adipogenic differentiation as assessed by lipid accumulation (ORO staining). The rescue of *Kap1* expression was achieved by introducing an shRNA-resistant allele. The expression of endogenous and exogenous *Kap1* (endo-*Kap1* and exo-*Kap1*, respectively), as well as adipogenic marker genes was assessed by qPCR. $n = 3$ biologically independent experiments. **$p < 0.01$, *$p < 0.05$, *t* test, compared with *shControl*. **e** Effect of knocking out *Kap1* on primary MEF adipogenic gene expression as assessed by qPCR. Lentivirus encoding Cre recombinase or control vector were introduced to *Kap1*loxp/loxp MEF cells to generate *Kap1* KO and control cells. KAP1 protein levels are shown on the right. $n = 3$ biologically independent experiments. **$p < 0.01$, *$p < 0.05$, *t* test. All panels: error bars—SD. Source data are provided as a Source Data file

KAP1 ChIP replicates (Supplementary Fig. 5B), confirming the quality of our data. Unlike ZFP30, which showed more abundant binding at day 2, KAP1's enrichment at these regions was not strongly modulated by differentiation (Fig. 5a, b; Supplementary Figure 5C and Supplementary Data 4). Consistent with previous findings[46,52], KAP1 binding localized more often at genes coding for KZFPs (Fig. 5c; Supplementary Fig. 5D) and at the 3′ end of genes (Supplementary Fig. 5E), compared with other co-regulators/TFs, such as p300 and ZEB1. For instance, over 20% of genes bound by KAP1 encode KZFPs (Supplementary Fig. 5F), while this is the case for only less than 2% of P300 or ZEB1-bound genes ($p < 10^{-16}$, Fisher's exact test). These results further validate the quality of our ChIP-seq data.

Interestingly, our data revealed that only a small subset (14, ~3%) of ZFP30-bound regions are also targeted by KAP1. While most of these regions were gene-proximal, a single location stood out as proximal to a gene showing a consistent transcriptional response upon *Zfp30*'s absence in both 3T3-L1 and IBA cells (Fig. 3c): the *Pparg* locus. Strikingly, KAP1 binding mirrored ZFP30 binding at the *Pparg2* enhancer site, with D2-specific enrichment for both proteins (Fig. 5d). We found that multiple active histone marks (including H3K27ac and H3K4me1), transcriptional coactivators (CBP/p300), as well as RNA Pol II were also only detected at this location upon adipogenic differentiation (Fig. 5d). We note that this was the case in both white and brown fat cell lines[38,39,53], suggesting that this regulatory element has pan-adipogenic specificity (Fig. 5d).

Our current understanding of KZFP–KAP1 interactions would suggest that KAP1 is recruited to the *Pparg2* locus by ZFP30. We therefore performed ChIP-qPCR with the KAP1 antibody in *Zfp30* KD and control cells before and after adipogenic differentiation. Consistent with the ChIP-seq data (Fig. 5d), the binding of KAP1 increased during differentiation, and this effect was reduced after *Zfp30* KD (Fig. 5e). In contrast, the binding of KAP1 to the *Zfp810* genomic locus, which is not bound by ZFP30, was not influenced by a ZFP30 loss of function (Fig. 5e). Moreover, we found that the recruitment of KAP1 to this genomic locus is impaired in *Zfp30* KO cells reconstituted with the KAP1-interaction-deficient mutant In24aa (Fig. 4c) compared with those of wild-type *Zfp30* (Supplementary Fig. 5G, H), excluding the possibility that ZFP30 binds/opens this locus and as such allows KAP1 to be recruited by other factors. Altogether, these data indicate that KAP1 is recruited to the *Pparg2* enhancer by ZFP30.

We next tested the function of KAP1 in regulating *Pparg2* expression. Since knocking down *Kap1* led to cell death in 3T3-L1 but not IBA cells (Supplementary Fig. 4), and KAP1 occupancy at the *Pparg2* enhancer was similar in the two cell lines (Supplementary Fig. 5I), we used IBA cells. *Kap1* KD diminished *Pparg2* enhancer activity, which was reversed by restoring *Kap1* expression (Fig. 5f), demonstrating the direct involvement of KAP1 in regulating *Pparg2* expression. Moreover, ectopic expression of *Pparg2* restored the differentiation defect, both in *Kap1* and *Zfp30* KD IBA cells (Fig. 5g, h). Together, these results

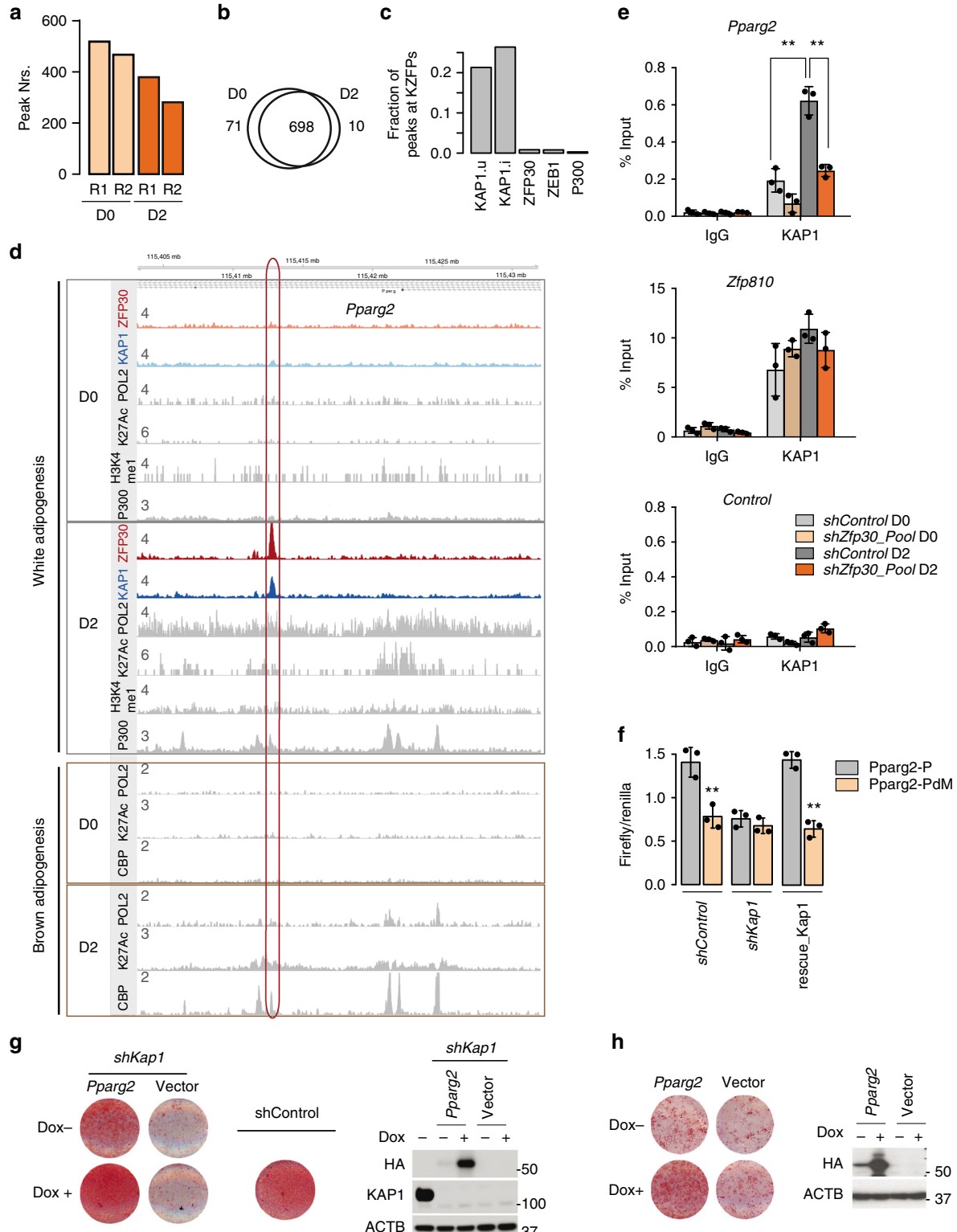

support our thesis that ZFP30/KAP1 promotes adipogenesis by directly targeting *Pparg2*.

## Discussion

Most of our knowledge on fat cell differentiation is derived from studies employing the 3T3-L1 and 3T3-F442A cell models. While many regulators identified from these models have proven to

be relevant outside these systems, others still need to be confirmed. For example, p38 promotes adipogenesis in 3T3-L1 cells[54], but shows an opposite effect in other models[55]. Our study reveals that the poorly characterized KZFP ZFP30[56,57] is important for adipogenesis across different mouse pre-adipocyte cell lines, an in vivo mouse model and ex vivo human cells, firmly establishing it as a pro-adipogenic TF. ZFP30 directly enhances *Pparg2* expression through binding and recruiting KAP1 to a

**Fig. 5** ZFP30 activates *Pparg2* expression through KAP1. **a** Number of KAP1 ChIP-seq peaks in 3T3-L1 cells at days 0 and 2 upon adipogenesis. Two biological replicates (R1 and R2) were performed. **b** The overlap between KAP1 ChIP-seq peaks at days 0 and 2. **c** The fraction of distinct categories of peaks (union of all KAP1 – KAP1.u, KAP1 present in at least two samples – KAP1.i, ZFP30, ZEB1, or p300) proximal to a gene encoding a KZFP. **d** KAP1, ZFP30-HA, RNA PolII (POL2), p300/CBP, H3K4me1, and H3K27ac (K27Ac) ChIP-seq enrichments at the *Pparg2* locus in 3T3-L1 cells (upper panel, gray rectangle) or brown pre-adipocytes (lower panel, brown rectangle) at days 0 and 2 of adipogenic differentiation. **e** ChIP qPCR probing IgG or KAP1 in 3T3-L1 cells upon *Zfp30* KD or shRNA control at days 0 and 2. The KAP1-bound *Pparg* and *Zfp810* loci as well as a negative control were tested. $n = 3$ biologically independent experiments. **$p < 0.01$, $t$ test. **f** Luciferase assay showing that KAP1 is essential for the identified *Pparg2* enhancer activity. Transcriptional activity as measured by luciferase assays (Firefly/Renilla) for Pparg2-P and Pparg2-PdM (see Fig. 3g) at days 2 of adipogenic differentiation upon *Kap1* KD and rescue, as shown in Fig. 4d. $n = 3$ biologically independent experiments. **$p < 0.01$, $t$ test. **g, h** Rescue of the differentiation defect in *Kap1* KD and *Zfp30* KO IBA cells, respectively, by ectopic expression of *Pparg2*. *Pparg2* expression can be induced by Doxycycline (Dox). Differentiation was assessed by lipid accumulation (ORO staining) while the protein levels were measured by western blotting. Note that there is already leaky expression of *Pparg2* when Dox is absent. All panels: error bars–SD. Source data are provided as a Source Data file

retrotransposon-derived enhancer located 9 kb upstream of the *Pparg2* TSS. *Pparg1* may also be regulated by this element, since its expression is reduced approximately fourfold in cells in which this enhancer is partially deleted (Supplementary Fig. 3G). However, the fact that *Pparg2* shows a ~50-fold expression reduction in these same cells suggests that *Pparg2* is the primary regulatory target (Fig. 3k ; Supplementary Fig. 3G). Consistently, *Pparg2* ectopic expression is sufficient to rescue the differentiation defect in ZFP30- or KAP1-depleted cells (Fig. 5g, h). While our current study focuses on ZFP30's role in adipogenesis and the relation with KAP1 and *Pparg2* expression, ZFP30's function likely goes beyond a single tissue, as exemplified by its link to disease phenotypes (*Cerebral visual impairment and intellectual disability*, ClinVar)[58], and its high expression in tissues, such as the retina, skin, and muscle (Supplementary Fig. 1I). For instance, the single experimentally documented role of ZFP30 prior to our study was its involvement in inflammation in the murine lung[56]. Our gene expression data align with this finding, suggesting that ZFP30 represses genes that enrich for inflammatory and immune response-related functionalities (Fig. 2d–f; Supplementary Data 2). Meanwhile, our results suggest that some genes can be both activated and suppressed by ZFP30, depending on their cellular context (Fig. 3d).

In our previous screen[20], we identified several zinc finger proteins, including ZEB1, ZFP30, and ZFP277 as top hits. Although all three of them belong to the same TF family, they appear to regulate adipogenesis through different mechanisms. Indeed, we found that ZFP277-HA did not even localize to the cell nucleus, while ZEB1 has a very broad target landscape compared with a restricted one for ZFP30. The most coherent regulatory adipogenic signal for ZFP30 pointed clearly to direct *Pparg* regulation. Even though *Zfp30* is already highly expressed before differentiation, it binds and activates *Pparg2* only after differentiation. This suggests that ZFP30 is not a pioneer factor in mediating chromatin access, and that it therefore requires other TFs to open the chromatin. Moreover, there are several adipogenic TF-binding motifs co-enriched in ZFP30-binding sites (Supplementary Data 5) and ZFP30 shares a large fraction of binding loci with these factors, which for example include the adipogenic pioneer factor C/EBPbeta and ZEB1[20,38,59] at the *Pparg* locus (Supplementary Fig. 2R). Based on these collective results, we propose that ZFP30 is a highly specific but integral part of the wider adipogenic gene regulatory network: its binding is facilitated by prior chromatin opening by C/EBPbeta or other factors, while it itself recruits KAP1 to increase *Pparg2* expression.

Indeed, as a member of the KZFP family, ZFP30 was expected to act through recruitment of the KAP1-silencing complex[49]. The latter contains the HP1, SETDB1, and NuRD complexes, as well as DNMTs, and mediates chromatin inactivation and/or DNA methylation. This silencing mechanism was found to be important for controlling transposable element (TE) activity and

the expression of nearby genes during the early stages of embryonic development[11–14] as well as in some adult tissues[18,19]. An evolutionary *arms-race* model was proposed to explain the constant competition between KZFPs and their TE targets[16]: KZFPs evolve to suppress new TEs, while TEs mutate to evade the repression, triggering further KZFP evolution. This model may explain the large amount and diversity of TEs in the genome, with about 44 and 38% of repetitive DNA in humans and mice, respectively[60], as well as the abundance of KZFPs. Indeed, a recent large-scale ChIP-seq study showed that most KZFPs bind TEs[46]. Consistently, we found that ZFP30 preferentially binds to specific subtypes of LINE1 elements, directly targeting L1MEd and L1MEg elements in particular (Fig. 3a, b), across mice and humans.

However, the arms-race model only partially explains the evolution of KZFPs and their DNA interaction properties, because most TE-associated KZFP-binding sequences are far more conserved than the rest of the elements in which they reside, and some KZFPs emerged long after their TE had lost any invading potential[46]. Supporting a regulatory rather than a purely inhibitory role for KZFPs, many TE-residing KZFP-binding sites are enriched for active enhancer marks in a cell type-specific manner[46]. This suggests that over time, certain TE-bound KZFPs were adopted into the local gene regulatory network to mediate transcription, similar to other TE-targeting TFs, such as the master genome regulator CTCF[61]. Here, we propose that ZFP30 constitutes a prime example of such a possible exaptation process[18]: ZFP30 targets a degenerated LINE-1 (L1MC5a) element and increases the expression of the nearby gene, the master adipogenic regulator *Pparg2*. The ZFP30 target site is highly conserved across ~100 million years of evolution (Fig. 3e) and contains the newly described ZFP30 DNA-binding motif, which is required for *Pparg2* activation and bound by ZFP30 with high affinity in vitro. Interestingly, both ZFP30 itself and the transposable element subtypes that it primarily targets (L1MEd and L1MEg) have been found to stem from overlapping time windows, between 105 and 150 million years ago[45,46]. This suggests that ZFP30 may have initially evolved to suppress these elements. At the *Pparg2* locus, its binding to the L1MC5a insert could have evolved into an adipogenic hotspot[44], also targeted and activated by GR and C/EBPbeta[62], which forms a loop to interact with and activate the *Pparg2* promoter[63]. Whether and how ZFP30/KAP1 function in coordination with the previously identified hotspot factors remains to be fully elucidated.

Interestingly and as already mentioned above, we did find that similar to many, though not all KZFPs, ZFP30 binds KAP1. However, unlike all other KAP1-interacting KZFPs characterized so far, ZFP30 recruits KAP1 as a co-activator at the *Pparg2* enhancer (Fig. 4–5). While KAP1 has previously been shown to be a co-activator of non-KZFP TFs, such as C/EBPbeta[64], NGFI-B[65], and MYOD[66], our study shows that the KZFP–KAP1 axis can also function in a non-repressive manner. Future research

will need to uncover the exact molecular mechanism that allows KAP1 to function as a co-activator in this adipogenic context.

Finally, it has recently been shown that KAP1 haploinsufficiency triggers bistable obesity in mice, where the obesity state is characterized by a reduced expression of an imprinted gene network, including *Nnat*, *Plagl1*, and *Peg3*[67]. However, it is not clear whether this phenotype is a direct effect of KAP1 on adipose tissue or not. In fact, only *Peg3* among these three genes is differentially expressed in *Zfp30* KD 3T3-L1 cells (Supplementary Data 1). Further, the conditional KO of KAP1 in mature adipocytes (*Adipoq*-Cre) does not impact adiposity[67], indicating that KAP1 is not essential in fully differentiated adipocytes. However, whether KAP1 impacts adipocyte differentiation was not known. Here, we provide evidence that KAP1 is required for the differentiation program in pre-adipocytes, and that one of its functions depends on recruitment by ZFP30. Together, our findings provide a new understanding of both adipogenic and KZFP–KAP1-mediated gene regulation.

## Methods

**Cell culture**. 3T3-L1 (ATCC), 293T (ATCC), and parental IBA cells[32] were cultured in the high-glucose DMEM medium (Life Technologies, Carlsbad, CA) supplemented with 10% fetal bovine serum (FBS) (Gibco), and 1× penicillin/streptomycin solution (Life Technologies) in a 5% $CO_2$ humidified atmosphere at 37 °C and maintained at less than 80% confluence before passaging. Primary $KAP1^{loxp/loxp}$ MEF was generated from embryos and cultured in the same condition, as mentioned above.

**Generation of knockout IBA cells**. *Zfp30* KO IBA cells were generated using the CRISPR/Cas9 technology[29,30]. The targeting sequence was: 5′-GTGCCTGAGTGCCTACGAGA-GGG (the last GGG is the PAM sequence)-3′. Two days after transfection of the sgRNA and hCas9-Blasticidin (kindly provided by Dr. Jiahuai Han) bearing plasmids, the cells were selected with 8 µg/ml Blasticidin S HCl (#A1113903, Gibco) for 3 days. The selected cells were plated on 96-well plates in a series of dilutions to achieve a single cell per well. The grown-up clones were screened for *Zfp30* KO and WT by genomic PCR followed by DNA PAGE electrophoresis. The sequences of the screening primers were: TGGCCGTGGTGTCTCTTCTCA and GAGGTGACCATTCTAAACCACAAAG. Clones showing no wild-type band and those showing only wild-type bands were then chosen and the PCR products were subjected to TA cloning (K202020, Invitrogen). Ten TA clones from each IBA clones were subjected to Sanger sequencing. Among them, four clones carrying missense and frame-shift mutations, but not the wild-type allele, were identified as *Zfp30* KO; and five clones carrying only the wild-type allele were identified as *Zfp30* WT. We did not find any cells carrying more than two mutated alleles among all the cells that we analyzed, which shows that IBA cells are a diploid, at least at the *Zfp30* genomic locus.

The IBA cells containing a ZFP30-binding motif deletion in the focal Pparg enhancer were generated by the same approach with some modifications. Two sgRNA sequences (sg1: TTCAATAATCATTCTGCTTA-GGG and sg2: CTTCTCCAAATAAATGATTT-GGG) were cloned into PX458 (Addgene #48138), respectively. IBA cells were transfected with these two plasmids together, and the GFP + cells were single-cell sorted into 96-well plates 48 h post transfection. Afterward, the downstream procedure is the same as detailed above for generating Zfp30 KO cells. The sequences of the screening primers were: AGAGAAATGCCAACTGAGTGA and TCCTCTTCAAGAACAGTGGGT. Candidate clones were further validated by TA cloning and Sanger sequencing. The resulting *Pparg2*-M1 KO clone used in this study contains a 43-bp deletion. To explore whether, next to ZFP30, other adipogenic TFs may possibly bind to this fragment (and additional 10 bp in each end), we performed a PWM scan for adipogenesis-related (GO:0003700 DNA-binding transcription factor activity and containing *lipid* or *fat* terms in their GO annotation) TFs derived from the mouse motif database HOCOMOCO (34 TFs, Supplementary Data 7)[68] and the vertebrae database JASPAR (33 TFs, Supplementary Data 7)[69] using FIMO (default settings)[70], revealing no significant hits.

**Generation of the *Zfp30*[tm3e] mice**. *Zfp30*[tm3e] targeted ES cells were purchased from the European Conditional Mouse Mutagenesis Program[34]. These cells were injected into B6 Albinos blastocysts, and chimeric male mice were bred to B6 Albinos female mice. The offspring carrying the *Zfp30*[tm3e] allele were identified by genotyping and bred to C57BL/6J mice. The genotyping primers are listed below:

Zfp30-tm2a-qF: GTTCACAGTGCCAAATTCTGG
Zfp30-tm2a-qR: AGGTGGAGTTCAGGGTCAG
Neo-F: CCTGAATGAACTGCAGGACG
Neo-R: GTTTCGCTTGGTGGTCGAAT

The *Zfp30*[tm3e] allele shows a band at 250 bp and wild-type allele at 125 bp. The whole targeting cassette was validated by long-range PCR using SequalPrep Long PCR kit (Applied Biosystems) followed by Sanger sequencing.

**Human SVF cells**. The human SVF cells were extracted with a previously described method[36]. Briefly, 50 ml of fresh lipoaspirates were washed twice with 40 ml of DPBS Ca+/Mg+(Gibco #14040091) in 100-ml syringes (VWR International #720-2528) and subsequently incubated with 0.28 U/ml of liberase (Roche #05401119001 (ROC)) for 45 min at 37 °C under agitation. The digested tissue was mixed with 40 ml of 1% human albumin (CSL Behring) in DPBS Ca–/Mg– (Gibco #14190094) and shaken to liberate the stromal cells. The aqueous phase was recovered and centrifuged at 400 g for 5 min at RT. The cell pellet was resuspended in 15 ml of remaining buffer and filtered through a 100-µm and then 40-µm cell strainer. The cell suspension was centrifuged and resuspended in 5 ml of 5% human albumin (CSL Behring). The viability and the number of nucleated cells in the obtained cells suspension were determined using a Nucleostainer, after which a red blood cell lysis was performed using VersaLyse solution (Beckman Coulter #A09777) according to the producer's recommendations.

**RNA interference and rescue**. The pLKO.1-based shRNA was purchased from Sigma.

>mouse shZfp30_1 (sigma: TRCN0000084321)
CCTGTTAGTCTCTATCAGAAA > mouse shZfp30_2 (sigma: TRCN0000084322)
CGTTGGGAAATAATGGGAAGAA > mouse shZfp30_3 (sigma: TRCN0000084318)
GCCATGATAGACTGGCTTGTA > mouse shKap1 (Sigma: TRCN0000071363)
CCGCATGTTCAAACAGTTCAA > human shZFP30-1 (sigma: TRCN0000107910)
CCTCCTAATGTGCTACAGTAA > human shZFP30-2 (sigma: TRCN0000107911)
GCCTTTAGAGTTAGAGGACAC > human shZFP30-3 (sigma: TRCN0000107912)
CGTCGTTACTCAGAACTTATT > human shZFP30-4 (sigma: TRCN0000107913)
CGACAACAACTTACTTTCCAT > human shZFP30-5 (sigma: TRCN0000107914)
AGTACGACAACAACTTACTTC

The lentivirus vector for KAP1 KD and rescue was described previously[66]. They derived from the pSicoR-Ef1a-mCh (Addgene #31847). Hairpin sequence against murine Kap1 (5′-CCGGCCGCATGTTCAAACAGTTCAACTCGAGTTGAACTG TTTGAACATGCGGTTTTTG-3′) or firefly luciferase (fLuc) were cloned in the HpaI/XhoI sites for creating the shControl and shKap1 lentiviral vectors, respectively. For the rescue-Kap1 plasmid, a Kap1 CDS containing a codon-optimized shRNA-resistant was cloned in the shKap1-bearing pSicoR-Ef1a-mCh vector to replace mCherry CDS. The rescue-ZFP30 (human) plasmids with human shZFP30-1 were generated in the same manner.

**Lentivirus production and infection**. Pre-confluent HEK293T cells were transfected with lentiviral vectors carrying cDNAs or shRNAs of interest and second generation of lentivirus-packing plasmids by the calcium phosphate precipitation method. The cell culture medium was changed in 12 h, and the virus-containing medium was collected 48 h post transfection.

For infection, a 1:1 ratio of virus-containing medium and fresh medium with Polybrene (final concentration at 10 µg/ml) were added to target cells at the density ~30–50%. The plates were centrifuged at 1500 g for 30 min and returned to the cell incubator. The infectious medium was changed 12 h later. A mock infection without a virus was included in each experiment.

**Adipogenic differentiation**. 3T3-L1 or IBA cells were induced to differentiation at day 0 (2 days post confluence) by adding the induction medium, which is the complete culture medium supplemented with the MDI cocktail (1 µM dexamethasone, 0.5 mM 3-isobutyl-1-methylxanthine, and 167 nM insulin, all from Sigma, Saint Louis, MO). At day 2, the induction medium was removed and the maintenance medium (the complete culture medium supplemented with 167 nM insulin) was added. At day 4, the maintenance medium was removed, and complete culture medium was added. The cells were harvested for staining or RNA extraction at day 6.

The isolated human SVF cells were then plated and cultured with MEM alpha (Gibco #32561037) supplemented with 5% human platelet serum and 50 µg/ml Primocin (InvivoGen #ant-pm-1). Once confluent, the cells were subjected to differentiation by exposing to the induction medium (the high glucose DMEM, 10% FBS, 50 µg/ml Primocin, 0.5 mM 3-isobutyl-1-methylxanthine, 1 µM dexamethasone, 1.7 µM insulin) for 7 days and subsequently the maintenance medium (the high glucose DMEM, 10% FBS, 50 µg/ml Primocin, 1.7 µM insulin) for another 7 days.

**Lipid staining**. Fixed cell lipid staining was performed using ORO. In total, 30 ml of 0.3% of the ORO stock solution (in isopropanol) was added to 20 ml of deionized water. The solution was filtered with filter paper and used as a working solution. Cells were washed with PBS twice and fixed with 10% formalin at room temperature for 30 min. Then they are washed with 60% isopropanol once, and stained with the ORO working solution for 10 min. The solution was poured off, and the cells were rinsed with water until the water run cleared. The cells were then scanned by image scanner or imaged under a microscope. For quantification, images were split into red, green, and blue channels by the *Split Channels* function in Fiji. The lipid accumulation was scored by the intensity of red channel subtracted from the average of those from green and blue channels.

Live cell lipid staining was performed using BODIPY 493/503 (D3922, Invitrogen). BODIPY and Hoechst were added directly to the cell culture medium for 15 min with final concentration at 1 µg/ml and 5 µg/ml, respectively. Then the medium was changed with the fresh medium, and the cells were imaged under a Leica DMI4000. The lipid (green channel) and nucleus (blue channel) intensity of human SVF cells were quantified by ImageJ/Fiji.

**RNA extraction and qPCR**. RNA extractions were performed using the Direct-zol RNA Miniprep kit (R2052, Zymo Research). Potential genomic DNA contamination was eliminated by DNase I digestion. The RNA quality and concentration was checked by Nanodrop spectrophotometer (Thermo Scientific).

One microgram of total RNA was used for single-strand cDNA synthesis using the SuperScript VILO cDNA Synthesis kit (Life Technologies). The cDNA was diluted 40 times, and 2.5 µl was used for qPCR. qPCR reactions were assembled using PowerUp SYBR Green Master Mix (ThermoFisher Scientific) with three technical replicates. The qPCR was performed using the QuantStudio 6 Flex qPCR system (Applied Biosystems). Primers were checked for linearity and single-product amplification. The sequences of the primers are listed below:

Mouse Zfp30-1: forward-CATGGCTCATAGTACGGTG; reverse-TACAGAT CCCTCTCGTAGG

Mouse Zfp30-2: forward-AAGCCAGATGTGATCACCCT; reverse-GCCACG CCTTTGAATTCTTC

Mouse Pparg2: forward-CACCAGTGTGAATTACAGCAAATC; reverse-AGC TGATTCCGAAGTTGGTG

Mouse AdipoQ: forward-TGGGGACCACAATGGACTCTA; reverse-TGGGC TATGGGTAGTTGCAGT

Mouse Hprt1: forward-ATACCTAATCATTATGCCGAGG; reverse-AGCAAG TCTTTCAGTCCTG

Mouse Fabp4: forward-CATCAGCGTAAATGGGGATT; reverse-GTCGTCTG CGGTGATTTCAT

Human ZFP30: forward-CAAGCCATTCTCCTCCCTCA; reverse-ATGAAGT GAGAGTCGGGCAT

Human ADIPOQ: forward-CCATACCAGAGGGTAAGAGCA; reverse-CCAG CAACAGCATCCTGAG

Human HPRT1: forward- CGAGCAAGACGTTCAGTCCT; reverse-TGACCT TGATTTATTTTGCATACC

To quantify the transcript derived from the ZFP30-bound L1MC5a in the *Pparg* locus, a similar procedure was applied, except that random hexamers were used for reverse transcription. The primers are the same as those used to perform ChIP-qPCR at this *Pparg* locus and as listed in the ChIP-qPCR Methods section.

Zfp30-2 primers were used to perform qPCR on *ZFP30^tm3e* mouse tissues and qPCRs shown in the left panel of Supplementary Fig. 1D. Zfp30-1 primers (one primer matches the junction of the CRISPR/Cas9 targeting site, thus is sensitive to the KO alleles) were used for all the other qPCR experiments.

**In vivo adipogenesis assay**. The in vivo experiments were performed as described previously[20] by using the shRNAs listed above. Briefly, fat tissue was dissected, minced, and incubated in collagenase type II for 1 h at 37 °C. Approximately 10^6 cells were treated with virus and resuspended in 100 µl of Matrigel (BD Biosciences, San Jose, CA) before injection subcutaneously into a skin fold of the neck. After 6 weeks of high-fat diet, Matrigel pads were excised. From each pad, pictures of three full sections were taken, and adipocyte numbers as well as the number of nuclei were determined automatically using the Cell Profiler Software. All experiments were performed in three replicates, and the significance of the observed changes was estimated using a one-sided Wilcoxon rank-sum test.

**mRNA-seq**. The total RNA was checked for quality by Fragment Analyzer (Advanced Analytical). High-quality samples (RQN > 8) were used to prepare the library using the TruSeq RNA Library Prep Kit v2 (Illumina). Libraries were checked for quality and quantified using the Fragment Analyzer and Qubit (ThermoFisher Scientific), respectively, before being sequenced on Nextseq 500 (Gene expression core facility, EPFL). Two biological replicates were performed for each condition.

**mRNA-seq analysis and TE expression analysis**. Fastq files containing paired-end (ZFP30 KD and control samples in 3T3-L1 cells) and single-end (ZFP30 KO and control samples in IBA cells) sequenced tags (reads) from two replicate

experiments each, were analyzed. Reads from each sample were trimmed, filtered, and checked for quality using trim_galore 0.4.0 (cutadapt and FastQC)[71] with the parameters *-q 20 --fastqc --stringency 3 -e 0.05 --length 36*. The retained tags were aligned to the Ensembl 84 gene annotation of the NCBI38/mm10 mouse genome using Bowtie 0.12.9 and the parameters *-a --best --strata -S -m 100*. Expression levels per transcript and gene were estimated using mmseq-1.0.8[72] with default parameters. Quantile normalized expression estimates were transformed into pseudo-counts by un-logging, un-standardizing, and multiplying with gene length. Pseudocounts were then normalized using mean-variance modeling at the observational level, as implemented in the voom() function in limma_3.30.4[73]. Differential expression was computed on the normalized values using the limma_3.30.4 pipeline at an FDR of 0.1 and fold-change cutoff of 2 (|FC|). Heatmaps displaying row-normalized (z-score transformation) expression values were generated using the function heatmap.2() in gplots_3.0.1. Averages of the normalized expression values (log2, before the *z*-score transformation) are displayed as an additional sidebar left of the respective heatmaps.

For the TE analysis, reads (paired-end, 2 × 100 bp) were mapped to the mouse (mm10) genome using hisat2 with parameters hisat2 -k 5 --seed 42 -p 7[74]. Counts on genes and TEs were generated using featureCounts[75]. To avoid read assignment ambiguity between genes and TEs, a gtf file containing both was provided to featureCounts. For repetitive sequences, an in-house curated version of the Repbase database was used (fragmented LTR and internal segments belonging to a single integrant were merged). Only uniquely mapped reads were used for the quantification. Finally, the features for which the add up of reads across all samples was lower than the number of samples were discarded from the analysis. Normalization for sequencing depth was done for both genes and TEs using the TMM method as implemented in limma[73] and using the counts on genes as library size. Differential gene expression analysis was performed using limma voom[73]. A TE was considered to be differentially expressed when the fold-change between groups was > 2 and the *p*-value < 0.05. A moderated *t* test (as implemented in R limma) was used to test significance. *P*-values were corrected for multiple testing using the Benjamini–Hochberg's method[76].

**Chromatin immunoprecipitation (ChIP)**. Chromatin immunoprecipitation was performed as previously described[77]. Briefly, 2 × 15 cm dishes of cells were cross-linked with 1% formaldehyde at room temperature for 10 min. The cross-linking was quenched by fresh glycine at a final concentration of 0.15 M. The cells were then harvested and lysed in a series of ice-cold lysis buffer to collect the nuclei. One mililiter of nuclei suspension was transferred to Covaris 1 -mL glass tube and sonicated at 10% duty factor, peak incident power 140 W, and cycle per purst 200 for 20 min in Covaris E220. In total, 20 µl of the sonicated chromatin was de-cross-linked and the DNA was extracted for quality control. The majority of the DNA is ~200 bp. Those that passed the quality control were divided into 25 µg of DNA in each aliquot and 1 µl was kept as input. Overall, 75 µl of magnetic protein G beads coupled with 10 µg of anti-HA (ab9110, Abcam), anti-KAP1 (KAP1-a1: ab10483 (Abcam), KAP1-a2: ab22553 (Abcam)), anti-p300 (sc-585, Santa Cruz), or anti-IgG control antibody (12-371, Millipore) was incubated with each aliquot of chromatin overnight at 4 °C. The beads were then washed and subjected to different processing listed below:

ChIP-qPCR: the chromatin precipitated by the beads, together with the input chromatin were digested with Proteinase K and RNase, then subjected to de-cross-linking by heating at 65 °C overnight. The DNA was then purified using the Agencourt AMPure XP (Beckman Coulter). qPCR was performed as mentioned before. Note that the ChIP-ed DNA was normalized to the input DNA, leading to the *%input* value. The primers are listed below:

*Pparg*: forward-TCTTCGTGTTATAAACCCAAATCA; reverse-GGGTTCTAA AACCTTCATCATCTC

*Zfp810*: forward-CTGACCTACACTGGAATGCC; reverse-TCACAGGACTCA TACAGGTGA

Control: forward-GGGTGCTAGCCTTCCTGACT; reverse-TCCAAGGTTCTC CCGACATA

*Rasgfr1*: forward-CTGCTGCTCCCACATCCA; reverse-GCAGTCGTGGTAGT TGTA

*Zfp316*: forward-TTCATGTGCGTGAGGAGG; reverse-AAGCGTTTCTCTCA GCGA

*Grb10*: forward-CATACGTGTTACATGCGC; reverse-TGTCGGTTCGTTTAG GAG

ChIP-seq: the ChIPmentation method[78] was used to generate the ChIP-seq library. In brief, the chromatin precipitated by the beads was subjected to Tn5 transposase-mediated tagmentation and adapter ligation at one single step. Then the tagmented chromatin was digested with Proteinase K and RNase, then de-cross-linked by heating at 65 °C overnight. The DNA was then purified using the Agencourt AMPure XP beads, and amplified using the KAPA HiFi Hotstart ReadyMix kit (KAPA Biosystems) and Nextera Index (Illumina) primer sets for 15 cycles. The PCR product was size-selected using Agencourt AMPure XP beads, quality checked using Fragment analyzer, quantified by Qubit, and then be sequenced on Nextseq500 (Gene expression core facility, EPFL). Two biological replicates were applied to each condition.

ChIP-exo experiments of human ZFP30 in HEK293T were performed as previously described[46]. In brief, ChIP was performed using 15 μg of anti-HA.11 antibody (clone 16B12, Covance) and protein G dynabeads (ThermoFisher). ChIP-exo was then performed on bead-bound chromatin, with adaptations for use in 96-well plates and Illumina sequencing. The original protocol was followed with the following modifications: we used protein G coupled to magnetic beads instead of sepharose; we doubled the amount of washes, but used only 200 μl instead of 1 ml; we used Ampure-based instead of column-based cleanup and size selection steps.

**ChIP-seq analysis.** The ChIP-seq tags were aligned to the NCBI38/mm10 genome with Bowtie2[79] and the parameters '--very-sensitive -p 8'. Duplicates were removed, as well as the reads with a mapping quality under 10. Regions showing significant ChIP enrichment (peaks) were determined with Homer[42] ('-F 2 -L 3 -localSize 5000 -C 4 -fragLength 200 -minDist 500 -center') using matched IgG (for ZFP30-HA and KAP1) and input (for p300, available from[20]) samples as controls. ZFP277-HA was additionally employed as a negative control based on its lack of localization to the nucleus, as previously described[20]: all regions overlapping ZFP277-HA peaks were removed from the analysis. Given the large variability in peak numbers between replicates at individual days despite high correlations of ChIP enrichments between the replicates (Supplementary Fig. 2E) and the fact that often, peaks called in a single replicate at day 0 were also detected in at least one replicate at day 2, we merged ZFP30 and KAP1-bound regions into either the union of all regions (union, u, Figs. 2h, 5b) or the intersection of at least two regions, irrespective of the day (in2, I, Supplementary Figs. 2H, 5C), respectively. When comparing differences in binding at the 2 days, we considered both peak numbers but also took a more quantitative approach, in which read counts contained in the genomic intervals defined by ZFP30 or KAP1 binding, respectively, at day 0 to day 2 were compared using DESeq2 1.14.1[80] with $padj \leq 0.15$, $|FC| > 2$. Shared and D0 or D2-specific peak numbers were displayed using the R package VennDiagrams 1.6.17. Library-size normalized ChIP-seq tags were visualized using Gviz 1.18.2 (Figs. 2i, 5d; Supplementary Figs. 2I, 5D). Log2-transformed library-size normalized DNAse I and H3K4me1 ChIP-seq tags in a 4 -kb large region centered around the point of maximal ZFP30-binding (peak summit) were averaged and displayed in Fig. 2j (for differential ZFP30 peaks) and Supplementary Fig. 2J (constitutive ZFP30 peaks).

**Motif analysis.** For both mouse and human, motif discovery was performed using MEME 4.12.0 as part of the MEME-ChIP suite[70] ('-nmotifs 3 -minw 15 -maxw 45 –dna -revcomp) on 400 bp centered on the summits of all ZFP30-bound regions. In mouse, the set containing the intersection of at least two regions, irrespective of the day, was used (in2). ROC curves were calculated similarly to ref. [41]. Briefly, we considered a set of **p**, positive sequences of length **l** (ZFP30 peak sets, union) and a set of **n** negative sequences of length **l** (shifted ZFP30 sequences). All sequences were scored for the motif using Fimo (in MEME-ChIP suite) with the parameters --output-pthresh 0.1 and considering only the best-scoring motif hit/sequence, thus obtaining two vectors of **p** and **n** scores. The statistic **w** of a Wilcoxon-test between both vectors was computed and used to display the ROC curves in Supplementary Fig. 2N and compute the AUC as **w** /(**n** x **m**). The fraction of peaks with motifs displayed in Supplementary Fig. 2O corresponds to motif hits obtained using Fimo and the parameters—output -pthresh 0.001. To determine which TFs potentially co-bind with ZFP30, we expanded our motif analysis using HOMER findMotifs with the parameters $mopt = $ "-len 6,8,10,12,14,16 -size 100", reporting known motifs in Supplementary Data 5.

**Other computational analyses.** All computational analyses were performed using R version 3.3.2 and Bioconductor version 3.4. All t tests were unpaired if not otherwise specified. ZFP30-bound regions and regulated genes were annotated using Ensembl 84, either by direct download or through biomaRt_2.30.0[81]. The Gene Ontology enrichment analysis was performed using the topGO 2.26.0 package, the *elimCount* method and a *p*-value cutoff of 0.001. Repeat elements were obtained by scanning with RepeatMasker version 4.0.5 with the parameters -s -nolow -norna -species mammal -xsmall -html -gff. Repeat enrichment was obtained by comparing the fraction of ZFP30 peaks vs. shifted regions (same length and number) overlapping (1 bp) repetitive elements.

**Publicly available data.** We used the following publicly available data: (1) in 3T3-L1 cells: ChIP-seq using antibodies against RNA PolII, H3K4me1, H3K27ac, ZEB1, p300, C/EBPbeta, C/EBPalpha, RXR, PBX1, VDR, KLF5, ATF2, FOSL2, JUND, C/EBPalpha, STAT5A, cJUN, KLF4, ATF2, ATF7, GR, PAPRG, STAT1, and DNase-seq;[20,38,39,43,44] note that one replicate of H3K27ac data (day 2) is excluded from the analysis due to low quality. (2) In a brown pre-adipogenic cell line: RNA PolII, H3K27ac, and CBP[53]. Except for Supplementary Fig. 2R, where the original reported peaks were used, ChIP-seq and DNase-seq data were reanalyzed analogous to the in-house generated data.

**MITOMI.** The zinc finger domains of ZFP30 (amino acids 177–548) were cloned into pF3A–eGFP in vitro expression vector by Gateway cloning. The protein was expressed in vitro using 12 μl of the TNT SP6 High-Yield Wheat Germ protein expression system (Promega) and 4 μl of plasmid with concentration 200 ng/μl.

In total, 1 μl of the reaction was mixed with 4 μl of target DNA and loaded into MITOMI chamber. All target DNA fragments were obtained as single-stranded oligonucleotides listed below:

ChIP-seq consensus:
AGACACAGCTGAAGAGAGAATTAGAGAAATGGAAGA
  motif at PPARg: GGAGAAGAGATGATGAAGGTTTTAGAACCC
  control: ATTTGAGGGACCAACGCATGCAAGTCTTACTTTTAA

Each oligonucleotide contained a common 3′-prime end: CGTATGCCGTCT TCTGCTTG. These oligonucleotides were subsequently used to generate fluorescently labeled double-stranded DNA as described previously[40] with Cy5-labeling primer: Cy5-CAAGCAGAAGACGGCATACG.

The molds for microfluidic devices and devices themselves were adapted from MITOMI[40] as described previously[41] with minor modifications: the design did not contain capillary pumps for loading DNA and protein samples into MITOMI chambers. Instead, the samples were loaded by pressurizing the corresponding inlets of microchips. The designed microchips were fabricated using two-layer soft lithography, as described before[40]. The surface chemistry, MITOMI, and image acquisition were performed as described previously[40]. We quantified the amount of each target sequence bound to the respective TF at the equilibrium state by means of fluorescence in a range of input DNA concentrations in at least three replicates. The difference between the Cy5 signal in and outside MITOMI button normalized by GFP signal (corresponding to the amount of immobilized protein) was calculated for each concentration. The obtained binding curves for each sequence were then fitted by use of one-site equilibrium binding equation.

**Luciferase reporter assay.** Exo III-assisted ligase-free cloning method[82] was used to clone the Zfp30-binding sequence of the targets into pGL3-promotor vector (Promega), between the NheI and BglII restriction sites. The primers to amplify the target sequences are listed below:

*Tec*: forward-cttacgcgtgctagcGGTGACTGAGAAGAAGCCCA; reverse-gcagatc gcagatctGGTCGTCAAAGCAATGCCTT

*Plscr2*: forward-cttacgcgtgctagcCGTGAGCTCGTTCTTCTTGG; reverse-gcagat cgcagatctTGCTCCTCGTTTCCAGTTCT

*Myof*: forward-cttacgcgtgctagcCTCGTGGAGAAAGCAAGTCG; reverse-gcaga tcgcagatctTCGAGAATAGTGGACAGCCC

*Col28a1*: forward-cttacgcgtgctagcTTCAGCCCTGCAAGTGATTG; reverse-gcag atcgcagatctTCCCTTCCCCTTTCCCTTTC

*Ahnak*: forward-cttacgcgtgctagcGGCCAGCACAGAAAGACTTC; reverse-gcag atcgcagatctTGGAACTGCAGCTGGTTGTA

*Pparg*: forward-cttacgcgtgctagcATTTGAGGGACCAACGCATG; reverse-gcaga tcgcagatctTGGGGTCAATTCTCAAGCCT

*Angptl4*: forward-cttacgcgtgctagcCACCTAAAGCCTACCCCACA; reverse-gca gatcgcagatctCCCTTCTTGACATCTGTGGC

*Cidec*: forward-cttacgcgtgctagcTCCAACAGGTGAGCAGACAT; reverse- gcaga tcgcagatctCAGCCCGGGTACTTCTCTAC

To generate the Pparg2-P luciferase reporter, 9.5 kb of DNA upstream the Pparg2 TTS was cloned. The Pparg2-PdM and Pparg2-PdP were generated by mutagenesis from the Pparg2-P fragment. All the DNA fragments were cloned into the pGL3-basic vector (Promega), using the XhoI and HindIII restriction sites. The primers are listed below:

For *Pparg2*-P:
*Pparg*-F1: tagcccgggctcgagCCAGTAGGAACTGCATTTCAGAGT
*Pparg*-R1: cggaatgccaagcttGAATCTCCCAGAGTTTCACCCATAAC

For *Pparg2*-PdM fragment1:
*Pparg*-F1: shown above
mutation-R: TCCAAATAAATGATTTGGGTTTATAACACGAAGA

For *Pparg2*-PdM fragment2:
Mutation-F: ATAAACCCAAATCATTTATTTGGACCCTAAGCAGAATGAT TATTGAAACTGAG
*Pparg*-R1: shown above

For *Pparg2*-PdP:
*Pparg*-F2: tagcccgggctcgagCATGAGTGTCACAGTAACTGCACA
*Pparg*-R1: shown above

Note that the bases in lowercase are the overhang sequences used to anneal with the vector.

**Immunoprecipitation and western blotting.** Cells were lysed with lysis buffer (20 mM Tris-HCl, pH 7.5, 150 mM NaCl, 1 mM Na2EDTA, 1 mM EGTA, 1% TritonX-100, 2.5 mM sodium pyrophosphate, 1 mM -glycerophosphate, 1 mM Na3VO4) on ice for 15 min. Cell lysates were then centrifuged at 20,000 *g* for 30 min. The supernatant was immunoprecipitated with 20 μl of mouse anti-HA-agarose beads (A2095, Sigma) at 4 °C for 3 h or overnight. After the immuno-precipitation, the beads were washed four times in lysis buffer, and the immuno-precipitated proteins were subsequently eluted by 1× SDS sample buffer at room temperature for 10 min. The supernatant was collected and denatured at 100 °C for

2 min. The samples were subjected to western blotting with the corresponding antibodies. Rabbit anti-Flag (F7425, dilution 1:1000), mouse anti-α-tubulin (T5168, 1: 5000), and mouse anti-HA antibodies (H3663, dilution 1:1000) were purchased from Sigma. Rabbit anti-HA (ab9110, dilution 1:1000) and mouse anti-KAP1 (ab22553, dilution 1:1000) were from Abcam. The uncropped western blot images are provided in Supplementary Figure 6.

**Bioethics**. All mouse experiments were conducted in strict accordance with Swiss law, and all experiments were approved by the ethics commission of the state veterinary office (60/2012, 43/2011). The work on human SVF cultures derived from human lipoaspirate samples is approved by the ethical commission of Canton Ticino (CE 2961 from 22.10.2015) and conforms with the guidelines of the 2000 Helsinki declaration. The anonymized samples were collected under signed informed consent.

**Reporting summary**. Further information on research design is available in the Nature Research Reporting Summary linked to this article.

## Data availability

All RNA-seq and ChIP-seq data are available in the ArrayExpress database (www.ebi.ac.uk/arrayexpress) under the accession numbers E-MTAB-6995 and E-MTAB-6817. The uncropped western blots are provided in Supplementary Figure 6. The source data underlying Figs. 1a-b, 1d, 1f, 1h-k, 2l, 3d, 3f, 3h-i, 3k, 4d-e, 5e-f, and Supplementary Figs 1a-b, 1d, 1f-g, 1i, 1l, 2a, 2g, 3d-e, 3g, 5a, 5h-i are provided as Source Data file. All other relevant data supporting the key findings of this study are available within the article and its Supplementary Information files or from the corresponding author upon reasonable request. A reporting summary for this article is available as a Supplementary Information file.

## Code availability

The codes used for the analysis are incorporated into the Methods sections listed above.

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

## Acknowledgements

We thank Pricilla Turelli and Romain Groux for constructive discussions. We thank Magda Zachara for careful reading of the manuscript. We thank Zoltán Kutalik for his help with interpreting human obesity-related genomic data. We thank Carla Rudigier for experimental support. We thank Maria Litovchenko for her help with motif analysis. We thank Jiahuai Han for kindly providing the lentivirus-packing plasmids. This research was supported by the Human Frontier Science Program LT001032/2013 (to PCS), by European Union's Horizon 2020 Framework Programme for Research and Innovation under grant agreement 665667 (to WC), by the Swiss National Science Foundation Grants (#31003A_138323, 31003A_162735, 31003A_162887, and 310030_182655) and by institutional support from the Swiss Federal Institute of Technology in Lausanne (EPFL) and in Zürich (ETHZ). We thank the EPFL BIOP, and Gene Expression Core Facilities for respectively imaging and sequencing support, as well as the VITAL-IT platform (University of Lausanne) for computational support.

## Author contributions

B.D., W.C., and P.C.S. designed the study and wrote the paper. W.C., E.V.P., C.G., S.K.R., R.D., M.I., J.R., and T.D. performed the experiments. P.C.S. analyzed all the next-generation sequencing-related data and W.C. all other data. J.D. performed data analysis of TE expression. M.C., S.M.J., and D.T. contributed unpublished essential reagents. D.T. and C.W. revised the paper.

## Additional information

**Competing interests:** The authors declare no competing interests.

