## [Peer Review File · Nature Communications]

Reviewers' comments:

Reviewer #1 (Remarks to the Author):

The manuscript describes how an enhancer element upstream of the promoter of *Pparg2* is bound by ZFP30 and KAP1, which promotes its activation and subsequent activation of *Pparg2* transcription. The most striking discovery is the activating role of KAP1/ZFP30, which are usually found with repressive functions.

Although the majority of data presented was acquired with mouse cell lines, this is a valid study of the regulation of *Pparg2*, a key gene for adipogenesis, which constitutes a obesity and type 2 diabetes GWAS locus.

The experimental perturbations on several adipogenic differentiation models show excellent attention to technical detail and are of high quality, even though most perturbations shown are still based on shRNA, which starts to be outdated given the availability of CRISPR/Cas9 for gene repression.

This study constitutes a follow up on previous art from the same authors, where ZFP30 was a top hit in a screening study aimed to identify pro-adipogenic factors. Although data on ZFP30/Kap1 role in adipogenesis and *Pparg2* regulation is convincing, major improvement is required regarding computational data analysis, which I enumerate below.

Major comments:

- In "We observed a striking reduction of lipid accumulation (as assessed by Oil Red O - ORO - staining), which was anti-correlated to the level of *Zfp30* expression (Figure 1A and Supplementary Fig. 1A)." There must be a typo, since lower expression of *Zfp30* seems to lead to less lipid accumulation. Thus correlation rather than anti-correlation is probably observed. Wording such as correlation should not be used without application of a statistical test to show it. Given that the data presented seems to relate so well, a statistical test for correlation between lipid accumulation and *Zfp30* and/or adipogenic marker expressed and *Zfp30* would actually help the authors make their point better. P values should be incorporated into the manuscript.

- To show that the IBA CRISPR/Cas9 KO lines are true KOs, the authors performed qPCR using a primer at the junction of the CRISPR cut site (legend for Fig. S1C says "One primer matches the junction of the CRISPR/Cas9 targeting site, thus is sensitive to the KO alleles."). This assay is misleading, suggesting at first sight that no *Zfp30* mRNA is being produced in KO clones, which is not being demonstrated if one of the primers does not anneal the mRNA of truncated isoforms depicted in Figure S1C. In fact, sequence information shown in Fig. S1C suggests that several alleles do not have frameshift mutations and may therefore express truncated forms of ZFP30. Specifically, clone 15 seems to resemble more a heterozygous clone, since one of its alleles contains a 24 amino acid insertion, but no premature stop codon or frameshift mutation. *Zfp30* KO should ideally be demonstrated by Western blot, but given the lack of available antibodies (stated elsewhere in the manuscript), the authors should design a qPCR assay that is able to amplify cDNA in the KO clones, to show nonsense mediated decay.

- The authors expressed a tagged ZFP30 in order to perform genome-wide mapping of ZFP30 binding sites. It is worth mentioning that the authors rightfully expressed the target protein up to endogenous levels only to avoid overexpression-derived artefacts, which is excellent. However, a few points should be improved in the analysis of this ChIP-seq experiment. The authors show the number of peaks per replicate (Fig. 2G), but this is not informative of correlation between replicates. Information in Figure 2G could be moved to supplementary as it is redundant with information shown in Fig. 2H. Pulldown reproducibility should be demonstrated by showing correlation of signal across the full genome and number of overlapping peaks for the replicates each condition, not only by showing a few examples (Fig. 2I and Fig. S2E). Comparison between days 0 and 2 (Figure 2H and S2D) is also not clear. Rather than showing all peaks in one figure and only a selection in a second figure, authors should show how the consistent peaks of day 0 overlap the consistent peaks of day 2.

- When analysing day 2-specific peaks, the authors observe that those regions gain chromatin accessibility from day 0 to day 2, consistent with the notion of those being dynamic chromatin regions. The authors also used previously published H3K4me1 ChIP-seq data to show that the same regions gain this regulatory element-associated chromatin mark from day 0 to day 2. When the authors state “active enhancer-specific H3K4me1 histone modification”, the sentence should be corrected for “enhancer-specific H3K4me1 histone modification” since this mark may also be present at primed enhancers that are not active yet. H3K27ac would be a better indicator of enhancer activity. In line with this analysis, to make the case that “overall chromatin structure of day 2-specifically bound regions may be more dynamic than that of other bound regions”, the authors should refer to Siersbæk et al Mol Cell 2017, where chromatin structure rewiring during adipogenesis is investigated.

- The authors then perform a series of analysis and experiments to identify the DNA binding motif of ZFP30 in mouse and human cells. While the approach is very thorough, the reasoning to switch to ChIP-exo, and not use conventional ChIP-seq, in human embryonic kidney cells, and not human adipocytes, is completely absent from the manuscript. While the authors state “we generated ChIP-exo data of ZFP30-HA in human HEK293T cells (Methods)”, the methods section only states “ChIP-exo experiments of human ZFP30 in HEK293T were performed as previously described (Imbeault et al. 2017).”

- A natural step after showing the analysis of differential expression for Zfp30 KD/KO and ChIP-seq for ZFP30 would've been to demonstrate that the binding of ZFP30 associates with direct changes in gene expression. How many of the down- and up-regulated genes have associated ZFP30 peaks? This analysis shown later in Figure S3A, but would make more sense in the context of Figure 2.

- When investigating the role of ZFP30 at L1-associated genes, the authors decide to focus only on genes proximal (< 5kb) or overlapping L1-associated ZFP30 peaks. In light of recent data using HiC and pChIC showing that regulatory elements may affect genes hundreds of kilobases away, why did the authors not investigate other potential targets, located within the same TAD, or with evidence of interaction of these peaks in adipocytes?

- The authors observed context-specific effects of ZFP30 KO/KD, detecting both up and downregulation of L1-proximal genes. This observation renders the question of which partners give specificity to ZFP30 action. Is there a DNA binding motif specific to either class of genes that could indicate a specific binding partner for activation/repression?

- The authors observe that “We found that while there were ≥ 30 genes bound and up- or down-regulated in KD or KO samples”. Is this more than expected by chance? A P value for the comparison against a randomized control set should be provided in the manuscript.

- The comment above also applies to the following quote: “Overall, genes responding to Zfp30 expression reduction were about two times more likely to be bound compared to all expressed genes, and genes repressed by ZFP30 at adipogenic day 2 were even more enriched for ZFP30 binding (Supplementary Fig. 3A)”. The analysis shown in this figure does not correspond to the statement. Figure S3A simply shows the fraction of differentially expressed genes per category, which does not represent enrichment over all expressed genes. Also not clear what the dashed line is meant to represent in Supplementary Fig. 3A.

- The authors deploy MITOMI to validate the binding of ZFP30 to the binding site detected by ChIP-seq near the TSS of Pparg2. Two points. 1) It would be more informative to see a read pileup of the HA-ZFP30 ChIP, rather than the position of the called peak in Figure 3E. 2) Why was it necessary to validate the ChIP-seq by MITOMI? Reasoning not clear and gives impression that ChIP-seq is not trustworthy.

- In Figure 3G-I the authors present an experiment showing that ZFP30 binding is important for the activation of the Pparg2-specific promoter. However, the authors seem to be over-inflating the results saying "enhancer activity was eliminated when Zfp30 was KD, indicating that this enhancer activity is indeed mediated by ZFP30". While it is true that the activity of the enhancer is diminished by ZFP30 KD, it is not completely abolished, being observed a significant difference between the empty vector control and the other conditions shown in Figure 3I. Similarly, Figure 3H shows that there is still enhancer activity above background level with constructs lacking ZFP30 motif. Thus the results the authors show clearly demonstrate the key role of ZFP30 in the regulation of this promoter, but also that ZFP30 is not the only key regulatory factor involved. Statistical analysis should be presented using the negative control (pGL3 vector) as control sample, and not the full construct.

- The authors then investigate the interaction between Kap1 and ZFP30 using elegant KD followed by rescue experiments (Figure 4C). Data clearly shows dramatic effect of Kap1 KO on adipogenesis.

- Similarly to points raised by ZFP30 ChIP-seq analysis, reproducibility of pulldown should be demonstrated with correlation of signal across the genome. Raw number of peaks (Fig. 5A) is redundant, given that intersection of peaks (Fig. 5B) contains this information. Peaks used for intersection between different time points should include only high confidence peaks, ie peaks present in both replicates, which cannot be the case given that the total number of peaks presented for day 2, for example, is over the maximum number of peaks per replicate of day 2. Regardless of which criteria was used to select regions shown in Figure 5B, it should be mentioned in the figure legend.

- The authors state that "KAP1 binding was highly enriched at genes encoding for KZFPs (Figure 5C, Supplementary Fig. 5C and Supplementary Table 5C) and at the 3' end of genes (Supplementary Fig. 5D)". Nevertheless, what is shown is the fraction of KAP1 peaks only. This does not show enrichment per se and should be compared to a random set. Comparison between 5' and 3' peaks also requires statistics to show that there is higher enrichment at 3' versus 5'. P values should be provided.

- The authors only focus on Pparg2 regarding the effects of the enhancer bound by Kap1/ZFP30. What is the evidence that the enhancer only regulates Pparg2 and not Pparg1 as well? Enhancers may regulate multiple genes simultaneously and at different distances. CRISPR and CRISPR activation could help resolve this question.

- Regarding the broader implications of this study, it would be interesting to know whether ZFP30/Kap1 are dysregulated in obese or obesity-associated diseases. Are there any GWAS variants mapping near either of these genes?

Minor comments:

- Although it is well known in the field that Pparg2 is indeed an isoform of the Pparg gene, this should be more explicit in the introduction and across the manuscript for readers that are not from the adipogenesis field. The authors seems to use both nomenclatures interchangeably. For example: authors say "the expression level of the adipogenic marker genes Pparg2, Adipoq and Fabp4 were significantly lower ($p < 0.01$, t-test) in Zfp30 knockdown (KD) cells compared to the control (Figure 1B).", even though the axis of the graph is labelled as Pparg. – which is true?

- In line with the comment above, authors should explain why it is important that the IBA line does not express Ucp1, as this is not common knowledge outside the adipogenesis field.

- When the authors say "The different differentiation capacity in KO and WT is not driven by the cell growth or proliferating rate, as the cell numbers are comparable (Supplemental Fig. 1F)", sentence could state more clearly when the measurements were made (after differentiation),

which is only present in the figure legend. Same issue when referring to Supplementary Fig. 1L.

- Figure S1G complements information in Figure 1D. What happens to Fabp4 in the other clones? Please add this information in Figure S1G.

- Small typo: please correct "stromal vascular faction" by stromal vascular fraction.

- Luciferase reporter assays shown in Figure 3D need a bit more explanation on what is being measured: are all DNA fragments distal elements, or are some of them be promoters? What is the negative control of the experiment? Are all DNA fragments of approximately the same size?

- Throughout the manuscript, the authors should only show union or intersection of peaks for analysis, since showing both is redundant.

- When the authors mention "We found that multiple active histone marks (including H3K27ac, H3K4me1), transcription coactivators (CBP/p300) as well as RNA Pol II were also only detected at this location upon adipogenic differentiation (Figure 5D).", H3K4me1 cannot be found in any panel of Figure 5 or S5.

- "KAP1 is indeed recruited to the Pparg2 enhancer by ZFP30." – Given the lack of antibody against ZFP30, the authors cannot perform a ChIP-on-ChIP experiment, which would indeed demonstrate direct recruitment of KAP1 by ZFP30. Therefore, the authors are advised to rephrase this sentence to "ZFP30 is indeed involved in the recruitment of KAP1 to the Pparg2 enhancer".

- Figure 6C: reference for the statistical test should be the negative control.

Reviewer #2 (Remarks to the Author):

Earlier work in the Deplancke lab identified ZFP30 in a forced-expression screen for TFs that increase adipogenic differentiation of 3T3-L1 cells. Here, Chen and colleagues follow up on that finding by investigating the molecular role played by ZFP30 in fat cell differentiation. They report that knockdown or CRISPR-mediated knockout of ZFP30 attenuates adipogenic differentiation in vitro or using implant models. Mechanistically, the authors find that ZFP30 targets LINE1 elements, and suggest that a complex of ZFP30 and KAP1 mediates transcriptional activation of Pparg2 from an ancient LINE element 9kb upstream of the Pparg2 transcription start site. This study builds upon the emerging concept that retrotransposons can be co-opted as cis-acting regulatory elements in the genome during the course of evolution, and provides an intriguing biological example of an unconventional role for KAP1 in mediating transcriptional activation as opposed to its classic role as a repressor. While interesting, several major concerns must be addressed to elevate enthusiasm for the study.

Major concerns:

1. The notion that ZFP30-dependent transcriptional activation of an intronic enhancer (retrotransposon 9kb upstream of the Pparg2 TSS) accounts for the ability of ZFP30 to regulate adipogenesis is not directly tested. This should be performed by CRISPR-mediated deletion of the ZFP30 motif at the genomic locus in IBA cells. The effect of genetic deletion of this putative enhancer on adipogenic differentiation is necessary to have confidence in the mechanistic conclusions of this study.

2. An appeal of this study is new insight into the genomic function of KZFPs, yet this is diminished by the inability to probe the genomic landscape for endogenous ZFP30 by ChIP-seq or ChIP-exo. It is admirable that the authors undertook the effort to produce cell lines that exogenously express HA-tagged ZFP30, but the genomics data lack important analyses and controls that provide insight into their robustness.

a. The relatively small number of binding sites identified for HA-tagged ZFP30 by ChIP-seq in mouse cells raises a red flag. Have the authors ruled out the possibility that the ChIP-seq peaks are due to off-target effects of the HA antibody? HA ChIP-seq in 3T3-L1 cells without exogenously expressed HA-tagged ZFP30 will address this.

b. The authors should report the hit rate (% of peaks with the motif) and p-value for their de novo ZFP30 motif in Figure 2K. Were other motifs co-enriched? These should also be reported, as it gives a relative measure of the enrichment for the top-ranked motif and may also provide insight into collaborating TFs.

c. CHIP-exo is a powerful approach, yet most of the analyses that are usually performed are not presented, and Figure 2M lacks important information. At the very minimum, the authors should show heatmaps and average profiles of the opposite-stranded peak pairs, and metrics including hit rate and p-value need to be included with the motif. Were multiple motifs identified? This information is required to judge the data quality. Moreover, It is concerning that the number binding sites for HA-tagged ZFP30 identified by CHIP-exo in human cells is 6-20-fold higher than that found by CHIP-seq in mouse cells. In general, the opposite result is expected, i.e. fewer binding sites for CHIP-exo versus CHIP-seq.

d. The preceding concerns raise questions about the genomic function of ZFP30. Investigating the adipogenic properties of a ZFP30 mutant lacking DNA-binding activity would help to alleviate these.

3. Given the moderate (mouse) to weak (human) differentiation defects in ZFP30 knockdown cells, more controls are needed to strengthen the conclusions of Figure 1. For example, the adipogenic effect of ZFP30 knockdown in human primary pre-adipocytes is small with differentiation changes of 20% or less for 4 of the 5 clones in Figures 1G and 1H. Moreover, the magnitude of ZFP30 knockdown and the degree of adipogenic impairment are not correlated well, especially when comparing clones 1 and 5, where the latter has no differentiation defect but the largest reduction in ZFP30 mRNA levels. How do the authors account for this result? Western blots showing ZFP30 protein in KD cell lines may provide an answer. In addition, an essential experiment missing from Figure 1 is a rescue of defective differentiation by shRNA-resistant ZFP30 or a closely related ZFP homolog. On a minor note, Figure 1 would be improved by labeling each panel with the relevant cell type.

Minor concerns:

4. Luciferase assays: In Figure 3D, the direction of change in luciferase reporter activity is incongruent between D0 and D2 for several genes (*Plscr2*, *Col28a1*, and *Cidec*). How do the authors account for these disparate responses? In Figure 3G, it would be useful to evaluate how much the *Pparg2* reporters changes upon co-transfection with a ZFP30 expression construct, especially in light of the earlier finding in the Gubelmann paper showing that forced expression of ZFP30 promotes adipogenesis.

5. Figure 6C requires western blots to demonstrate that the level of KAP1 protein is similar across various mutants. These data are necessary to distinguish between effects mediated by 473 phosphorylation and KAP1 expression differences.

6. Forced expression of several ZFPs promoted 3T3-L1 adipogenesis in the Gubelmann paper. Do these ZFPs share a common mechanism of action with ZFP30? The authors should discuss whether the biology of ZFP30 is likely to be distinct within this group.

Reviewer #3 (Remarks to the Author):

The manuscript by Chen et al provides molecular details of how a transcription factor (TF) called ZFP30 regulates adipogenic differentiation. ZFP30 belongs to a family of Krüppel-associated box Zinc Finger Proteins (KZFP), which are encoded by about 600 genes in the mouse genome. The canonical model suggests that KZFP bind specific DNA sequences within transposons and recruit

KAP1 co-repressor to downregulate the target retrotransposon. The manuscript describes an atypical mode of action of ZFP30 to make two novel claims: (i) ZFP30 recruits KAP1 to activate rather than repress one target retrotransposon. (ii) Since this retrotransposon is located within an enhancer element of PPARG2 locus, ZFP30 activates PPARG2 gene, and in turn promotes adipogenic differentiation.

The findings reported in this MS are interesting given the paucity of our understanding of the largest family of TF, especially in the regulatory context beyond retrotransposon silencing. However, both the claims made in the MS require further substantiation by performing some critical experiments. This is important since ZFP30 has already been identified as an adipogenic regulator by the authors' previous study (Gubelmann et al 2014). Thus the current MS is expected to identify detailed molecular mechanism by which ZFP30 regulates adipogenesis. The critical points that need authors' attention are summarised in the following:

(1) The claim that ZFP30 activates retroelement/ gene expression via KAP1 is rather circumstantial. First, the downregulation of ZFP30 targets upon ZFP30 loss-of-function is much less frequent than upregulation (Line 258). It appears that the major role of ZFP30 is to repress targets, and not to activate them. By calling this phenomenon "context-specific" (line 238) the authors ignore the broad function of ZFP30 in repressing genes to focus on a single location at which ZFP30 may activate the gene (line 349). This case appears more like an indirect effect/ an exception than a canonical function of this ZFP30. In such a scenario, it is imperative to explain why ZFP30 behaves dramatically opposite depending on the target locus. E.g. is there a co-occurring TF? Are these indirect effects? It is also important to show the effect of ZFP30 knock-down on the expression of retroelements (by re-analyzing RNAseq) which are completely ignored despite being central to authors' narrative. By RNAseq it will be difficult to distinguish different instances of retroelements in the genome, but unique reads in histone marks ChIPseq / unique primers in ChIP-qPCR can directly assess changes in expression of ZFP30-targeted retroelement instances in undifferentiated and differentiated cells. In line with this, quantitation of retroelement-neighborhood effects on gene expression should be done to explain what proportion of ZFP30 peak neighbours are up- or down-regulated and distance between the promoter and ChIP peak. Is there a CTCF site in between the two that explains why some neighbors are upregulated and others are downregulated? Cell-line-specific differences noted by the authors further complicate the simple interpretation that ZFP30 activates the target genes. An explanation/ analysis of why certain targets are regulated cell-line-specifically is required.

(2) The link between ZFP30 with PPARG2 regulation in the context of adipogenesis must be further strengthened. First, if ZFP30 regulates adipogenesis via PPARG2 as claimed by authors, ectopic expression of PPARG2 should rescue differentiation defect in ZFP30/ KAP1 KO cells. This is especially relevant as KAP1 KO causes cell death in one cell line (Line 314), suggesting PPARG2 misregulation as a narrow interpretation of loss-of-function of ZFP30/ KAP1. Second, an experiment that would demonstrate regulation of endogenous PPARG2 by the retroelement is to delete the minimum possible sequence spanning ZFP30 motif in the endogenous locus of PPARG2 enhancer. Most master regulators are themselves regulated in a complex manner by long-range interactions, which cannot be captured by enhancer-reporter constructs used by the authors. This experiment will also partially answer the criticism of indirect effect of ZFP30 loss-of-function.

(3) The contribution of KAP1 to the regulation by ZFP30 adds to the complexity of the model. Since this is an important and non-classical finding for future research and KAP1 has already been linked with another adipogenic regulator C/EBPbeta (PMID 9742105), KAP1-ZFP30 nexus in adipogenesis requires further clarifications. First, the reduction in KAP1 binding to PPARG2 locus upon ZFP30 knock-down can be simply explained by lack of open chromatin at this locus without ZFP30, and not direct recruitment of KAP1 by ZFP30 as claimed by authors. To directly demonstrate that ZFP30 recruits KAP1 to PPARG2 locus the author should utilize specific domain deletions that uncouple a KZFP from binding to KAP1. Rescue of ZFP30 knock-out cells by such mutants will be useful in showing a direct recruitment of KAP1. Second, the phosphorylation on KAP1 seems to be unlinked to the proposed mechanism, and appears only circumstantial. An increase in global phosphorylation of KAP1 would have effects on all KAP1 targets and not just PPARG2. Does ZFP30 bind more effectively to phospho-KAP1? Does ZFP30 binds targets more efficiently in presence of phospho-KAP1? While identity of the kinase is important, it will be beyond

this MS. Nonetheless, the importance of the phosphorylation in regulating PPARG2 may be indirect and independent of ZFP30. Efforts and experiments should be made to integrate phosphorylation data in the rest of the manuscript, else can be skipped altogether.

(4) Regulation and specificity of ZFP30: It is not clear in which tissues is ZFP30 expressed in comparison with adipogenic tissue. Also authors show that ZFP30 mRNA levels decrease upon adipogenic differentiation (Fig. S1D), not expected from a TF that directly regulates the differentiation. In the light of high levels of ZFP30 prior to differentiation, why should it activate PPARG2 gene only after the differentiation sets in, when in fact the levels of ZFP30 go down? If anything, ZFP30 may assist other TF such as C/EBPbeta/ delta in activating PPARG2. The DNase hypersensitivity data suggest that ZFP30 binds to many targets after differentiation as a consequence of chromatin opening by other pioneer TF than a primary cause of the differentiation process (Fig. 2I/ J). Hence the function of ZFP30 must be explained in the context of regulation by other known TF. After all, as the authors themselves claim that adipogenic regulation 'is one of the better characterized differentiation networks'. The study will be much better received if the function of ZFP30 is integrated in the known differentiation network of adipogenesis.

Minor points:

(1) The abstract states 'adipogenic exaptation'. This is overinterpretation of the data and not necessary. The 'switch' in function of ZFP30 from repressor to activator has no basis. Such ideas should be reduced to a couple of lines in discussion.

(2) Line 107: What is MDI cocktail? No full form is given in the main text.

(3) Fig. 3B: The color code of Day 0 and Day 2 in control and KO/ KD is confusing. Also Fig. 3D: ratio of control to shRNA is non-intuitive, as against the normal shRNA to control ratio (where upregulation is positive and downregulation is negative).

(4) Line 293: "Note that we define..." does not make sense in the context of the interpretations just before this line.

(5) Line 306: 3T3-L1 cell line is not a 'physiological condition'.

(6) Line 337: How do the 500 peaks of KAP1 compare with other cell types such as ESC? Thousands of peaks were reported in previous studies.

(7) Line 401: Typo error ZPP30.

(8) Line 472: How are Nnat, Plagl1 and Peg3 regulated under the conditions of Loss of function of ZFP30?

Rebuttal Letter for Manuscript NCOMMS-18-19456:

The reviewers stated that “the experimental perturbations on several adipogenic differentiation models show excellent attention to technical detail and are of high quality” and that as such the data “on ZFP30/Kap1 role in adipogenesis and *Pparg2* regulation is convincing” (reviewer #1); that the study “provides an intriguing biological example of an unconventional role for KAP1 in mediating transcriptional activation as opposed to its classic role as a repressor” (reviewer #2), and that “the findings reported in this MS are interesting given the paucity of our understanding of the largest family of TF, especially in the regulatory context beyond retrotransposon silencing” (reviewer #3). However, the reviewers also indicated that several issues would need to be addressed to improve the robustness of the presented findings. We would like to thank the reviewers for their constructive comments, which we have systematically addressed below. Together, this led to the following major additions to the manuscript:

1. To strengthen the finding that *ZFP30* is involved in human SVF differentiation, we performed a rescue experiment in *ZFP30* KO hSVF cells (Figure 1J-K) **(Reviewer #2)**
2. To further characterize the function of the ZFP30 binding motif in the *Pparg2* locus in its endogenous genomic context, we deleted this sequence in IBA cells and found that fat cell differentiation is impaired (Figure 3J-K and Supplementary Fig. 3G). **(Reviewers #1,2,3)**
3. We identified a KAP1 binding deficient mutant ZFP30-In24aa (Figure 4C). We found that this mutant fails to recruit KAP1 at the *Pparg2* locus, supporting the idea that KAP1 recruitment to this locus depends on ZFP30 (Supplementary Fig. 5G-H). **(Reviewer #3)**
4. We performed rescue experiment in *Zfp30* KO and *Kap1* KD cells by ectopically expressing *Pparg2*, strengthening the conclusion that *Pparg2* is the main target of the ZFP30/KAP1 complex in an adipogenic context (Figure 5G-H). **(Reviewers #1,3)**
5. We showed that the S473E mutant fails to activate the *Pparg2* reporter in *Zfp30* KO cells, indicating that the regulatory function of S473 phosphorylated KAP1 depends on ZFP30 (Figure 6E). **(Reviewer #3)**
6. We analyzed the correlation between the CHIP-seq data (Supplementary Fig. 2E and 5B), which further supported the overall high quality of our CHIP-seq data. To validate the genomic function of ZFP30 experimentally, we generated a DNA binding domain mutation form and found that it loses its ability to bind to all four of the tested loci (Supplementary Fig. 2F-G). **(Reviewers #1,2)**
7. To find potential co-regulators of ZFP30, we performed a motif co-enrichment analysis (Supplementary Table 5) and as such identified several factors that co-bind with ZFP30 (Supplementary Fig. 2R), enabling us to position ZFP30 in the adipogenic regulatory network (lines 536-545). **(Reviewers #1,2,3)**
8. We found that none of the observed, active TE's is directly regulated by ZFP30 (Supplementary Fig. S3A-B), suggesting that ZFP30's current function is not directly related to the repression of still active TEs. In addition, we found that the highly decayed ZFP30-bound TE linked to the *Pparg2* enhancer is transcribed, which is attenuated when *Zfp30* is KO (Supplementary Fig. S3D). These findings support our conclusion that ZFP30 activates the TE-derived enhancer in the *Pparg2* locus. **(Reviewer #3)**
9. Other more minor data additions or modifications were also made to Supplementary Fig. 1A right panel, Supplementary Fig. 1D left panel, Supplementary Fig. 1G right panel, Supplementary Fig. 2D, Supplementary Fig. 2P-Q, Supplementary Fig. 3E, Supplementary Fig. 6 and a Supplementary file (uncropped Western Blot images).

Reviewer #1 (Remarks to the Author):

The manuscript describes how an enhancer element upstream of the promoter of *Pparg2* is bound by ZFP30 and KAP1, which promotes its activation and subsequent activation of *Pparg2* transcription. The most striking discovery is the activating role of KAP1/ZFP30, which are usually found with repressive functions. Although the majority of data presented was acquired with mouse cell lines, this is a valid study of the regulation of *Pparg2*, a key gene for adipogenesis, which constitutes an obesity and type 2 diabetes GWAS locus. The experimental perturbations on several adipogenic differentiation models show excellent attention to technical detail and are of high quality, even though most perturbations shown are still based on shRNA, which starts to be outdated given the availability of CRISPR/Cas9 for gene repression. This study constitutes a follow up on previous art from the same authors, where ZFP30 was a top hit in a screening study aimed to identify pro-adipogenic factors. Although data on ZFP30/Kap1 role in adipogenesis and *Pparg2* regulation is convincing, major improvement is required regarding computational data analysis, which I enumerate below.

Major comments:

1. In “We observed a striking reduction of lipid accumulation (as assessed by Oil Red O - ORO - staining), which was anti-correlated to the level of *Zfp30* expression (Figure 1A and Supplementary Fig. 1A).” There must be a typo, since lower expression of *Zfp30* seems to lead to less lipid accumulation. Thus correlation rather than anti-correlation is probably observed. Wording such as correlation should not be used without application of a statistical test to show it. Given that the data presented seems to relate so well, a statistical test for correlation between lipid accumulation and *Zfp30* and/or adipogenic marker expressed and *Zfp30* would actually help the authors make their point better. P values should be incorporated into the manuscript.

Response: We agree with the reviewer that our statement was not clear as we meant to convey that “the greater the reduction, the lower *Zfp30* expression”, hence our use of the term “anti-correlated”. We have now corrected it in the revised manuscript (lines 96-99). Additionally, we quantified the lipid accumulation as suggested, demonstrating a significant positive (Pearson’s $r=0.94$, $p=0.017$) correlation with the *Zfp30* expression level (Supplementary Fig. 1A in the revised manuscript). Please note that the quantification method was added to the **Methods** section.

2. To show that the IBA CRISPR/Cas9 KO lines are true KOs, the authors performed qPCR using a primer at the junction of the CRISPR cut site (legend for Fig. S1C says “One primer matches the junction of the CRISPR/Cas9 targeting site, thus is sensitive to the KO alleles.”). This assay is misleading, suggesting at first sight that no *Zfp30* mRNA is being produced in KO clones, which is not being demonstrated if one of the primers does not anneal the mRNA of truncated isoforms depicted in Figure S1C. In fact, sequence information shown in Fig. S1C suggests that several alleles do not have frameshift mutations and may therefore express truncated forms of ZFP30. Specifically, clone 15 seems to resemble more a heterozygous clone, since one of its alleles contains a 24 amino acid insertion, but no premature stop codon or frameshift mutation. *Zfp30* KO should ideally be demonstrated by Western blot, but given the lack of available antibodies (stated elsewhere in the manuscript), the authors should design a qPCR assay that is able to amplify cDNA in the KO clones, to show nonsense mediated decay.

Response: To address the reviewer’s concern, we tested *Zfp30* mRNA levels in the KO clones. Instead of observing lower mRNA expression due to the expected NMD, we found that *Zfp30* is still expressed at variable levels in the KO clones (Supplementary Fig. 1D Left Panel in the revised manuscript). We hypothesize that this may be due to NMD bypassing, which has been previously observed (e.g. Carter et al., EMBO J, 1996; Romao et al., Blood, 2000 just to name a few). Despite this expression, our Sanger sequencing results (Supplementary Fig. 1C) support the notion that we

generated loss-of-function alleles since they produce largely truncated ZFP30 forms that lost the functional KRAB domain as well as the zinc finger domains. In clone 15, there is indeed a *Zfp30* allele coding a ZFP30 with a 24aa insertion in the KRAB domain. As shown in **Figure 4C** in the revised manuscript, this mutant loses its interaction capacity with KAP1. Taken together, our results strongly support the notion that *Zfp30* function is disrupted in the reported KO cells.

3. The authors expressed a tagged ZFP30 in order to perform genome-wide mapping of ZFP30 binding sites. It is worth mentioning that the authors rightfully expressed the target protein up to endogenous levels only to avoid overexpression-derived artefacts, which is excellent. However, a few points should be improved in the analysis of this ChIP-seq experiment. The authors show the number of peaks per replicate (Fig. 2G), but this is not informative of correlation between replicates. Information in Figure 2G could be moved to supplementary as it is redundant with information shown in Fig. 2H. Pulldown reproducibility should be demonstrated by showing correlation of signal across the full genome and number of overlapping peaks for the replicates each condition, not only by showing a few examples (Fig. 2I and Fig. S2E). Comparison between days 0 and 2 (Figure 2H and S2D) is also not clear. Rather than showing all peaks in one figure and only a selection in a second figure, authors should show how the consistent peaks of day 0 overlap the consistent peaks of day 2. **Response:** We have now included correlations between the ChIP enrichments of *Zfp30* as well as IgG control replicates and days (**Supplementary Fig. 2E** in revised manuscript). As expected, samples are hierarchically organized, first by ChIP-ed “factor” and second by sampling point (day 0 versus day 2) – the stronger correlation of the replicates confirms the quality of our data (**lines 205-208**).

Second, we now provide an overview of all peak and overlap numbers (**Supplementary Fig. 2D**). We note that while the correlation between the individual replicates is high, there is a ~2-fold difference in the number of called peaks – thus, taking the intersection per day would remove a large number of putatively true positive peaks. Given this, in our manuscript, we took (1) a lenient approach, considering the union of all peaks and (2) a more stringent approach, considering peaks that are overlapping in at least two samples, irrespective of days. Given the overall low number of ZFP30 peaks genome-wide, it is highly unlikely that a false positive peak would be present at two distinct days in a single replicate only. It is more likely that low enrichment peaks are simply missed in the lower enrichment replicates. Given these data characteristics, a statistical comparison of the differences in enrichment inside regions bound by ZFP30 is meaningful in this context, therefore we have included **Figure 2H** and **Supplementary Fig. 2H** (Fig. 2H and Fig.S2D in the previous manuscript). We note that overall, the two approaches (peak overlap and differential enrichment) are qualitatively in agreement, revealing much higher ZFP30 binding at day 2. We now clearly describe the rationale for the approach taken in the **Methods** section of the manuscript.

4. When analysing day 2-specific peaks, the authors observe that those regions gain chromatin accessibility from day 0 to day 2, consistent with the notion of those being dynamic chromatin regions. The authors also used previously published H3K4me1 ChIP-seq data to show that the same regions gain this regulatory element-associated chromatin mark from day 0 to day 2. When the authors state “active enhancer-specific H3K4me1 histone modification”, the sentence should be corrected for “enhancer-specific H3K4me1 histone modification” since this mark may also be present at primed enhancers that are not active yet. H3K27ac would be a better indicator of enhancer activity. In line with this analysis, to make the case that “overall chromatin structure of day 2-specifically bound regions may be more dynamic than that of other bound regions”, the authors should refer to Siersbæk et al Mol Cell 2017, where chromatin structure rewiring during adipogenesis is investigated.

Response: We have now corrected the text and added the reference according to the reviewer’s suggestion. With regards to the H3K27ac data, we note that, given that we observed low concordance between the two replicates of the publicly available data (as shown below) and given

that this was not the case for the DNase I and the H3K4me1 data, we decided to not include this additional analysis.

Figure Legend: Two replicates of H3K27ac signal in a 4 kb window centered around the point of maximal ZFP30 binding at locations bound by ZFP30 at day 2 only.

5. The authors then perform a series of analysis and experiments to identify the DNA binding motif of ZFP30 in mouse and human cells. While the approach is very thorough, the reasoning to switch to ChIP-exo, and not use conventional ChIP-seq, in human embryonic kidney cells, and not human adipocytes, is completely absent from the manuscript. While the authors state “we generated ChIP-exo data of ZFP30-HA in human HEK293T cells (Methods)”, the methods section only states “ChIP-exo experiments of human ZFP30 in HEK293T were performed as previously described (Imbeault et al. 2017).”

Response: We thank the reviewer for raising this valid point. The human primary stromal vascular fraction (SVF) cells used in the manuscript were derived from lipoaspirations. As we could only obtain a limited amount of such samples, we were unable to generate a sufficient number of SVF-derived cells to perform ChIP-exo experiments. TF binding motifs are rarely tissue-specific or even species-specific for orthologous TFs, however, and have typically been derived from *in vitro* assays. Thus, in order to uncover the ZFP30 DNA binding motif in human cells, we resorted to HEK293T cells as an easier-to-use experimental system. ChIP-exo was applied, as it typically allows a greater resolution in identifying transcription factor binding sites, which benefits overall motif discovery.

Consistent with the notion detailed above, the top motif found in human was highly consistent with that found in mouse cells, as shown in **Figure 2M**. We have now added this description (**line 253-257**) and the Method details in the revised manuscript.

6. A natural step after showing the analysis of differential expression for Zfp30 KD/KO and ChIP-seq for ZFP30 would've been to demonstrate that the binding of ZFP30 associates with direct changes in gene expression. How many of the down- and up-regulated genes have associated ZFP30 peaks? This analysis shown later in Figure S3A, but would make more sense in the context of Figure 2. **Response:** We have moved the **Supplementary Fig. 3A** to **Supplementary Fig. 2K** in the revised manuscript, as suggested. The main text is also modified accordingly (**lines 233-237**).

7. When investigating the role of ZFP30 at L1-associated genes, the authors decide to focus only on genes proximal (< 5kb) or overlapping L1-associated ZFP30 peaks. In light of recent data using HiC and pcHiC showing that regulatory elements may affect genes hundreds of kilobases away, why did the authors not investigate other potential targets, located within the same TAD, or with evidence of interaction of these peaks in adipocytes?

Response: We did the analysis to associate the L1-associated ZFP30 peaks to gene expression in TADs (data from Siersbæk, et al., Mol. Cell, 2017) as suggested. In line with what was shown for proximal genes (see above), these genes also showed both up- and down- regulation (shown below). However, the overlap between these genes with proximal-regulated genes is low (only 7 genes) due to the incomplete coverage of TADs in the genome, for example ZFP30's main target *Pparg* is not included in a TAD. We thus would favor not including this analysis in our manuscript.

Figure legend: Heatmap showing the DE genes located within the same TADs with ZFP30 binding peaks.

8. The authors observed context-specific effects of ZFP30 KO/KD, detecting both up and downregulation of L1-proximal genes. This observation renders the question of which partners give specificity to ZFP30 action. Is there a DNA binding motif specific to either class of genes that could indicate a specific binding partner for activation/repression?

Response: What confers the context-specificity of ZFP30 action is a highly interesting but at the same time very challenging question. While there are indeed some motifs enriched in ZFP30-bound regions (**Supplementary Table 5 and Supplementary Fig. 2R**), we cannot perform motif enrichment analyses contrasting UP/DOWN associated motifs, given the low number of peaks when constraining the analysis to each condition. For instance, when considering genes that are UP/DOWN regulated consistently in both 3T3-L1 and IBA cells + proximal/in the same TAD as an L1-associated ZFP30 peak, we only have 1 gene, respectively *Pparg* for proximal peaks & *Prkag2* for TAD-contained peaks.

9. The authors observe that “We found that while there were ≥ 30 genes bound and up- or down-regulated in KD or KO samples”. Is this more than expected by chance? A P value for the comparison against a randomized control set should be provided in the manuscript.

Response: We have now modified the text to clearly state that 79 genes are bound- and upregulated in at least one of the KD or KO samples and 12 in both of them. The randomized control (the grey dash line) is highlighted in **Supplementary Fig. 2K** (Supplementary Fig. 3A in the original manuscript) and corresponds to 60 for the “or” condition and 5 for the “and”. P-values compared to a randomized control (based on 1,000 repetitions) = 0.003 (KD or KO) and 0.004 (KD and KO), respectively. This information has now been included in the revised manuscript (**lines 306-310**).

10. The comment above also applies to the following quote: “Overall, genes responding to Zfp30 expression reduction were about two times more likely to be bound compared to all expressed genes, and genes repressed by ZFP30 at adipogenic day 2 were even more enriched for ZFP30 binding (Supplementary Fig. 3A)”. The analysis shown in this figure does not correspond to the statement. Figure S3A simply shows the fraction of differentially expressed genes per category, which does not represent enrichment over all expressed genes. Also not clear what the dashed line is meant to represent in Supplementary Fig. 3A.

Response: We have corrected the axis annotation in **Supplementary Fig. 2K** (Supplementary Fig. 3A in the previous manuscript) – “Fraction of peak-prox. DE genes”. For distinct peak categories (3 distinct panels), we describe the fraction of DE genes (also distinct categories, in bar labels) that have at least one proximal ZFP30 peak. The mean overlap value for a randomized control is included as the dashed line (background overlap).

11. The authors deploy MITOMI to validate the binding of ZFP30 to the binding site detected by ChIP-seq near the TSS of *Pparg2*. Two points. 1) It would be more informative to see a read pileup of the HA-ZFP30 ChIP, rather than the position of the called peak in Figure 3E. 2) Why was it necessary to validate the ChIP-seq by MITOMI? Reasoning not clear and gives impression that ChIP-seq is not trustworthy.

Response: 1) The read pileup of HA-ZFP30 was shown in **Figure 5D** in the previous manuscript. We now also include it in **Figure 3E** in the revised version. 2) While the ChIP-seq shows which DNA regions are bound by ZFP30 in cells, this binding can be either direct or indirect, through recruitment by other factors. Here, we used MITOMI to show that ZFP30 binds to the target DNA /motif directly *in vitro*. We modified the text accordingly to clarify this in the revised manuscript (**lines 246-251 and 340**).

12. In Figure 3G-I the authors present an experiment showing that ZFP30 binding is important for the activation of the *Pparg2*-specific promoter. However, the authors seem to be over-inflating the results saying “enhancer activity was eliminated when Zfp30 was KD, indicating that this enhancer

activity is indeed mediated by ZFP30". While it is true that the activity of the enhancer is diminished by ZFP30 KD, it is not completely abolished, being observed a significant difference between the empty vector control and the other conditions shown in Figure 3I. Similarly, Figure 3H shows that there is still enhancer activity above background level with constructs lacking ZFP30 motif. Thus the results the authors show clearly demonstrate the key role of ZFP30 in the regulation of this promoter, but also that ZFP30 is not the only key regulatory factor involved. Statistical analysis should be presented using the negative control (pGL3 vector) as control sample, and not the full construct.

Response: We agree with the reviewer that ZFP30 is not the only key regulatory factor involved in the activation of the whole upstream region from -9.5 kb to -1 bp. Many factors are reported to bind and regulate this region by binding to different motifs, such as C/EBP alpha and beta or RXR (Lee, et al., Cell Biosci., 2014). To specifically address the contribution of ZFP30 to the DNA region it binds, which is as short as a 24 bp fragment, we defined the activity of this small region by contrasting the *Pparg2-P* (full length) and *Pparg2-PdM* (*Zfp30* motif mutation) reporter activities. In **Figure 3I**, as pointed out by the reviewer, the *Pparg2-PdM* still shows enhancer activity, but there is no difference in the activity of the *Pparg2-P* and *Pparg2-PdM* reporters when *Zfp30* is knocked down, indicating that *Zfp30* is essential for this small piece of regulatory DNA, but not for the residual activity of the 9.5 kb to -1 bp region. To avoid any confusion, however, we have modified the discussed description in the revised manuscript to "the difference between the *Pparg2-P* and the *Pparg2-PdM*/*Pparg2-PdP* reporter activities disappeared when *Zfp30* mRNA levels were reduced by KD (**Figure 3I**)" (**lines 358-360**). As our main goal is to compare the difference between *Pparg2-P* and *Pparg2-PdM*, we believe that it is more reasonable to use the *Pparg2-P* as a statistical / baseline control.

13. The authors then investigate the interaction between Kap1 and ZFP30 using elegant KD followed by rescue experiments (Figure 4C). Data clearly shows dramatic effect of Kap1 KO on adipogenesis.

Response: Thank you.

14. Similarly to points raised by ZFP30 ChIP-seq analysis, reproducibility of pulldown should be demonstrated with correlation of signal across the genome. Raw number of peaks (Fig. 5A) is redundant, given that intersection of peaks (Fig. 5B) contains this information. Peaks used for intersection between different time points should include only high confidence peaks, ie peaks present in both replicates, which cannot be the case given that the total number of peaks presented for day 2, for example, is over the maximum number of peaks per replicate of day 2. Regardless of which criteria was used to select regions shown in Figure 5B, it should be mentioned in the figure legend.

Response: In **Figure 5A**, we show the peak numbers across days and replicates based on the peak calling. In **Figure 5B**, we show the overlap based on differential binding analysis (when comparing the raw signal across the union of peaks). We believe that both these plots are informative & valuable to get the full overview of the distribution of the peaks across days, as described above and now included in the **Methods** section. We have now added correlation values as well in **Supplemental Fig. 5B**. The high correlation between replicates confirms the quality of our data.

15. The authors state that "KAP1 binding was highly enriched at genes encoding for KZFPs (Figure 5C, Supplementary Fig. 5C and Supplementary Table 5C) and at the 3' end of genes (Supplementary Fig. 5D)". Nevertheless, what is shown is the fraction of KAP1 peaks only. This does not show enrichment per se and should be compared to a random set. Comparison between 5' and 3' peaks also requires statistics to show that there is higher enrichment at 3' versus 5'. P values should be provided.

Response: In **Figure 5C** and **Supplemental Fig. 5E,F** (Figure 5C, Supplementary Fig. 5D,E in the previous manuscript), we included three other proteins as controls rather than a randomized control, demonstrating that KAP1's binding pattern is highly specific compared to other TFs (ZFP30, ZEB1) or chromatin regulators (p300). For the 3' vs. 5' end comparison, what is striking is that while ZFP30,

ZEB1, and p300 show higher overlaps with 5' versus 3' ends, the opposite is true for KAP1, as displayed in **Supplementary Fig. 5D**. For instance, over 20% of genes bound by KAP1 encode KZFPs (**Supplementary Fig. 5F**), while this is the case for less than 2% of P300 or ZEB1-bound genes ($p < 10^{-16}$, Fisher's exact test). We have now rephrased the text to specify that we included these factors for comparison and also included the p value in the revised manuscript (**lines 422-430**).

16. The authors only focus on *Pparg2* regarding the effects of the enhancer bound by Kap1/ZFP30. What is the evidence that the enhancer only regulates *Pparg2* and not *Pparg1* as well? Enhancers may regulate multiple genes simultaneously and at different distances. CRISPR and CRISPR activation could help resolve this question.

Response: As suggested by the reviewer as well as other reviewers, we have now generated IBA cells lacking the ZFP30 binding site (43 bp including the motif and several surrounding nucleotides) in the *Pparg* enhancer using CRISPR (**Figure 3J**). As shown in **Figure 3K**, KO of the motif largely reduced the lipid accumulation in 4 out of 5 clones of IBA cells, compared to the 4 wildtype clones. *Pparg2* expression was reduced to ~2% on average (~50 fold change) in the 4 enhancer KO cells, and *Pparg1* also showed a reduction to ~24% on average (~4 fold-change) (**Figure 3K and supplementary Fig. 3G**). These novel data show that the enhancer indeed regulates both isoforms, but mainly *Pparg2* in an adipogenic context, which is further supported by other new data demonstrating that the ectopic expression of *Pparg2* is sufficient to rescue the differentiation defect in *Zfp30* KO or *Kap1* KD cells (**Figure 5G-H**). We have included these exciting data in the revised manuscript (**lines 360-369 and 460-462**).

17. Regarding the broader implications of this study, it would be interesting to know whether ZFP30/Kap1 are dysregulated in obese or obesity-associated diseases. Are there any GWAS variants mapping near either of these genes?

Response: We have looked up all SNPs within 40kb of these two genes for significant associations with obesity related traits from the UK Biobank. As shown below in the QQ-plot of the associations, while there is some enrichment, it is not particularly striking. Thus, while potentially interesting, we cannot draw a firm conclusion at this stage, which is why we favor not including this analysis in the revised manuscript.

Figure legend: QQ-plot showing the empirically observed association significance between SNPs in *ZFP30* and *KAP1* loci and obesity-associated traits (y-axis) as a function of significance values expected from a normal distribution with the same mean and variance as the empirical distribution (x-axis).

Minor comments:

- Although it is well known in the field that *Pparg2* is indeed an isoform of the *Pparg* gene, this should be more explicit in the introduction and across the manuscript for readers that are not from the adipogenesis field. The authors seems to use both nomenclatures interchangeably. For example: authors say “the expression level of the adipogenic marker genes *Pparg2*, *Adipoq* and *Fabp4* were significantly lower ($p < 0.01$, t-test) in *Zfp30* knockdown (KD) cells compared to the control (Figure 1B).”, even though the axis of the graph is labelled as *Pparg*. – which is true?

Response: We appreciate the importance of this distinction which is why we have now added an introduction to the two isoforms *Pparg1* and *Pparg2* in the revised manuscript (lines 68-70). We have also modified the text and figures to consistently use *Pparg1*, *Pparg2* when referring to the isoforms and only used *Pparg* when referring to the gene in general.

- In line with the comment above, authors should explain why it is important that the IBA line does not express *Ucp1*, as this is not common knowledge outside the adipogenesis field.

Response: We have added a brief introduction to *Ucp1* in the revised manuscript (lines 110-111).

- When the authors say “The different differentiation capacity in KO and WT is not driven by the cell growth or proliferating rate, as the cell numbers are comparable (Supplemental Fig. 1F)”, sentence could state more clearly when the measurements were made (after differentiation), which is only present in the figure legend. Same issue when referring to Supplementary Fig. 1L.

Response: We have added a description of the measurement conditions in the revised manuscript (lines 125 and 155).

- Figure S1G complements information in Figure 1D. What happens to *Fabp4* in the other clones? Please add this information in Figure S1G.

Response: We have added the qPCR data of *Fabp4* in **Supplementary Fig. 1G** in the revised version.

- Small typo: please correct “stromal vascular faction” by stromal vascular fraction.

Response: We have corrected the typo in the revised manuscript.

- Luciferase reporter assays shown in Figure 3D need a bit more explanation on what is being measured: are all DNA fragments distal elements, or are some of them be promoters? What is the negative control of the experiment? Are all DNA fragments of approximately the same size?

Response: The *Myof* fragment is located at the promoter and the others are either intronic or distal intergenic elements. The negative control is the luciferase vector pGL3-promoter, which all the reporters are derived from. All the DNA fragments cloned are ~ 700 bp. We have added this information in the text (lines 315-320) and respective figure legend of the revised manuscript.

- Throughout the manuscript, the authors should only show union or intersection of peaks for analysis, since showing both is redundant.

Response: As also explained above and indicated in the **Methods** section of the manuscript, we have now included a **Supplementary Fig. 2D** with an overview of all peaks, including unions and intersections thereof. Given the small ZFP30 peak number at day 0 and the relatively high difference in number of called peaks in the two replicates at day 2, we have decided to perform our analyses with both a more lenient (union) and stringent (intersection of at least two samples, any day, int2) threshold. This is because in the differential binding analysis, we have noticed, that some peaks that would potentially be missed if only the intersection of D0 would be taken, are captured by our int2 approach. We believe that it is valuable to still report results obtained with both of these two approaches, as they show a consistent trend in terms of D0 vs. D2 peaks or other overlaps.

- When the authors mention “We found that multiple active histone marks (including H3K27ac, H3K4me1), transcription coactivators (CBP/p300) as well as RNA Pol II were also only detected at this location upon adipogenic differentiation (Figure 5D).”, H3K4me1 cannot be found in any panel of Figure 5 or S5.

Response: We have now added H3K4me1 in **Figure 5D**. We have also updated the H3K27ac data in this figure, as we discovered that one replicate of these data is in fact of low quality (mentioned above in our response to comment #4), which is why we decided to exclude it from the analysis. We have now noted this in **lines 1306-1307**.

- “KAP1 is indeed recruited to the *Pparg2* enhancer by ZFP30.” – Given the lack of antibody against ZFP30, the authors cannot perform a CHIP-on-CHIP experiment, which would indeed demonstrate direct recruitment of KAP1 by ZFP30. Therefore, the authors are advised to rephrase this sentence to “ZFP30 is indeed involved in the recruitment of KAP1 to the *Pparg2* enhancer”.

Response: We have added new data in **Supplementary Fig. 5G-H**. It shows that a KAP1-interaction-deficient mutant (In24aa) (**Figure 4C**), which is derived from the allele 2 of *Zfp30* KO clone c15 (**Supplementary Fig. 1C**), fails to recruit KAP1 at the *Pparg2* locus while the wild type ZFP30 is able to do so. It further supports our thesis that KAP1 is recruited to the *Pparg2* enhancer by ZFP30 (**lines 448-452**).

- Figure 6C: reference for the statistical test should be the negative control.

Response: As the main goal is to compare the difference between the S473A/S473E mutant and the KAP1-WT, we believe that it is more intuitive to use the KAP1-WT as a control. We have now added a description in the figure legend of **Figure 6C** to make the reference control more visible in the revised manuscript.

Reviewer #2 (Remarks to the Author):

Earlier work in the Deplancke lab identified ZFP30 in a forced-expression screen for TFs that increase adipogenic differentiation of 3T3-L1 cells. Here, Chen and colleagues follow up on that finding by investigating the molecular role played by ZFP30 in fat cell differentiation. They report that knockdown or CRISPR-mediated knockout of ZFP30 attenuates adipogenic differentiation in vitro or using implant models. Mechanistically, the authors find that ZFP30 targets LINE1 elements, and suggest that a complex of ZFP30 and KAP1 mediates transcriptional activation of *Pparg2* from an ancient LINE element 9kb upstream of the *Pparg2* transcription start site. This study builds upon the emerging concept that retrotransposons can be co-opted as cis-acting regulatory elements in the genome during the course of evolution, and provides an intriguing biological example of an unconventional role for KAP1 in mediating transcriptional activation as opposed to its classic role as a repressor. While interesting, several major concerns must be addressed to elevate enthusiasm for the study.

Major concerns:

1. The notion that ZFP30-dependent transcriptional activation of an intronic enhancer (retrotransposon 9kb upstream of the *Pparg2* TSS) accounts for the ability of ZFP30 to regulate adipogenesis is not directly tested. This should be performed by CRISPR-mediated deletion of the ZFP30 motif at the genomic locus in IBA cells. The effect of genetic deletion of this putative enhancer on adipogenic differentiation is necessary to have confidence in the mechanistic conclusions of this study.

Response: Please see our response to reviewer 1 (major comment #16), detailing our efforts in

generating IBA cells lacking the ZFP30 binding site (43 bp including the motif and several surrounding nucleotides) in the *Pparg* enhancer using CRISPR and as is represented in **Figure 3J-K**.

2. An appeal of this study is new insight into the genomic function of KZFPs, yet this is diminished by the inability to probe the genomic landscape for endogenous ZFP30 by ChIP-seq or ChIP-exo. It is admirable that the authors undertook the effort to produce cell lines that exogenously express HA-tagged ZFP30, but the genomics data lack important analyses and controls that provide insight into their robustness.

a. The relatively small number of binding sites identified for HA-tagged ZFP30 by ChIP-seq in mouse cells raises a red flag. Have the authors ruled out the possibility that the ChIP-seq peaks are due to off-target effects of the HA antibody? HA ChIP-seq in 3T3-L1 cells without exogenously expressed HA-tagged ZFP30 will address this.

Response: Despite the low number of ZFP30-HA bound regions in the system, the characteristics of our ChIP-seq data (strong correlation between replicate experiments (**Supplementary Fig. 2E**), the determination of *in vitro* validated sequence-specificity based on ChIP-seq peak regions (**Figure 2L, 3F**) and our ample range of validation experiments all support the robustness of our HA-ZFP30 approach. In addition, in our previous manuscript (Gubelmann et al., eLife, 2014), we have found a large (instead of a highly restricted) set of ZEB1-HA bound regions using a similar approach. In both cases, we have directly used ChIP-seq of ZFP277-HA – a protein that is not localizing to the nucleus - as a negative control, in order to exclude off-target effects. We have now included this in the **Methods** section of the revised manuscript. More generally, it is not uncommon that multiple zinc finger domain-containing KZFPs (ZFP30 contains 13 zinc finger domains) have relatively few bound regions genome-wide, in particular considering their large DNA-binding motifs, e.g. ZFP568 (Yang et al., Science, 2017) and ZNF419 (Imbeault et al., Nature, 2017).

b. The authors should report the hit rate (% of peaks with the motif) and p-value for their de novo ZFP30 motif in Figure 2K. Were other motifs co-enriched? These should also be reported, as it gives a relative measure of the enrichment for the top-ranked motif and may also provide insight into collaborating TFs.

Response: We note that the information relevant to M1 (and the second hit, M2), was included in **Supplementary Fig. 2G-J** (**Supplementary Fig. 2L-O** in revised manuscript), including the motif E-values, motif-peak AUCs and fraction of peaks with motifs. We note that the third motif had an E-value of $6.7e-010$, an order of magnitude higher (thus less significant) than the reported M1 motif, therefore we decided not to include it in the manuscript.

That said, we have now analyzed the same peak regions using Homer, looking for enrichment of known motifs. As shown in **Supplementary Table 5**, several motifs were co-enriched. We have now incorporated these data in the context of ZFP30's position in the adipogenic regulatory network (**lines 537-542**) and see also the new **Supplementary Fig. 2R**.

c. ChIP-exo is a powerful approach, yet most of the analyses that are usually performed are not presented, and Figure 2M lacks important information. At the very minimum, the authors should show heatmaps and average profiles of the opposite-stranded peak pairs, and metrics including hit rate and p-value need to be included with the motif. Were multiple motifs identified? This information is required to judge the data quality. Moreover, It is concerning that the number binding sites for HA-tagged ZFP30 identified by ChIP-exo in human cells is 6-20-fold higher than that found by ChIP-seq in mouse cells. In general, the opposite result is expected, i.e. fewer binding sites for ChIP-exo versus ChIP-seq.

Response: The heatmaps and average profiles of the opposite-stranded peak pairs have now been included in **Supplementary Fig. 2P**. A similar analysis was also applied to the identified motif (**Supplementary Fig. 2Q**). The broad peaks likely reflect a suboptimal digestion by exonuclease.

However, both motif and ChIP-seq reads showed strong central enrichment at these bound regions, suggesting the high specificity of our data (**Supplementary Fig. 2P-Q**). The hit rate (64%) and E value (10^{-415}) have now also been mentioned in the text of the revised manuscript (**lines 262-265**).

While the ChIP-exo assay yields indeed in general fewer peaks than ChIP-seq experiments in the same cellular condition, it is difficult to compare the mouse ChIP-seq and human ChIP-exo data in our case for the following reasons: 1) 293T cells contain the large T antigen, which allows episomal amplification of plasmids containing the viral SV40 origins. Thus, it allows production of the recombinant protein at high levels by permitting the persistence of more plasmid copies in the transfected cells; 2) for the ChIP-seq of ZFP30 in 3T3-L1 cells, the mRNA expression of exogenous *Zfp30* was induced to a similar level as the endogenous *Zfp30* by adjusting the amount of Doxycycline, to avoid potential artefacts due to protein overexpression. This was mentioned in our main text: **lines 200-202**; 3) even in the same cell type, the peaks can vary in different conditions. For sample, we found a higher number of ZFP30-bound regions at day 2 of differentiation compared to day 0 in 3T3-L1 cells (**Figure 2G**). We hope that this clarifies the observed binding profiles.

d. The preceding concerns raise questions about the genomic function of ZFP30. Investigating the adipogenic properties of a ZFP30 mutant lacking DNA-binding activity would help to alleviate these.

Response: In response to the reviewer's request, we reconstituted the *Zfp30* KO IBA cells with wild-type as well as the DNA-binding zinc finger domain-deletion mutant (ZFP30-ΔZF), respectively (**Supplementary Fig. 2F** in the revised manuscript). We observed two bands corresponding to the ZFP30-ΔZF mutant in the blot, indicating the potential presence of a post translational modification. We then performed a ChIP experiment on these cells using the HA antibody and tested the enrichment of a few ZFP30 ChIP-seq-derived peaks by qPCR. As shown in **Supplementary Fig. 2G**, while we observed significant enrichment for WT ZFP30 at all four tested loci, the ZFP30-ΔZF mutant bound to a much lesser extent (indistinguishable from control). In contrast, the negative control was not targeted by ZFP30. Thus, these data, together with the old and novel data already discussed above, confirm the specificity of the ZFP30 ChIP-seq data and thus support the reported genomic function of ZFP30.

3. Given the moderate (mouse) to weak (human) differentiation defects in ZFP30 knockdown cells, more controls are needed to strengthen the conclusions of Figure 1. For example, the adipogenic effect of ZFP30 knockdown in human primary pre-adipocytes is small with differentiation changes of 20% or less for 4 of the 5 clones in Figures 1G and 1H. Moreover, the magnitude of ZFP30 knockdown and the degree of adipogenic impairment are not correlated well, especially when comparing clones 1 and 5, where the latter has no differentiation defect but the largest reduction in ZFP30 mRNA levels. How do the authors account for this result? Western blots showing ZFP30 protein in KD cell lines may provide an answer. In addition, an essential experiment missing from Figure 1 is a rescue of defective differentiation by shRNA-resistant ZFP30 or a closely related ZFP homolog. On a minor note, Figure 1 would be improved by labeling each panel with the relevant cell type.

Response: Both the shRNA-mediated knockdown using three distinct shRNAs and the CRISPR/Cas9-mediated knockout with 4 out of 5 clones, in two different preadipocyte lines and also in a transplanted mouse model, support ZFP30's function in mouse cells. However, as the reviewer points out, the magnitude of ZFP30 knockdown and the degree of adipogenic impairment are not correlated well in the human SVF cells. We therefore agree with the reviewer that a ZFP30 protein quantification by western blotting may provide additional clarity. For example, certain shRNAs may also act as miRNA-like molecules by suppressing expression at the level of translation, which would not be detected by qPCR measurements. This may be the case for shRNA-1, as it targets the 3'UTR of the *Zfp30* mRNA. However, as we mentioned in the previous manuscript, neither commercial ZFP30 antibodies nor four batches of customized ZFP30 antibodies successfully recognized endogenous

ZFP30. To strengthen the evidence supporting ZFP30's function in human cells, we have included the rescue experiment in human SVFs as suggested. Specifically, we have made a construct with both shRNA-1 (targeting the 3'UTR) and a shRNA-resistant ZFP30 (without the 3'UTR and also code-optimized) in the same vector in order to rescue the expression of ZFP30. As shown in **Figure 1J-K**, the introduction of the shRNA-resistant ZFP30 indeed rescues the differentiation, as assessed by lipid staining (BODIPY), of human SVFs impaired by *Zfp30* shRNA (**lines 155-161**). We have also labeled each panel of **Figure 1** with the relevant cell type as suggested.

Minor concerns:

4. Luciferase assays: In Figure 3D, the direction of change in luciferase reporter activity is incongruent between D0 and D2 for several genes (*Plscr2*, *Col28a1*, and *Cidec*). How do the authors account for these disparate responses? In Figure 3G, it would be useful to evaluate how much the *Pparg2* reporters changes upon co-transfection with a ZFP30 expression construct, especially in light of the earlier finding in the Gubelmann paper showing that forced expression of ZFP30 promotes adipogenesis.

Response: We thank the reviewer for raising this interesting concern. In the discussion section of our previous manuscript, we discussed that, based on the RNA-seq and ChIP-seq data, ZFP30 likely acts as a context-dependent activator and suppressor. Based on our luciferase results, these genes appear to be differentially regulated by ZFP30 between D0-undifferentiated and D2-differentiated cells. This suggests that distinct regulatory mechanisms involving ZFP30 may exist for the same gene in different cellular contexts. We agree with the reviewer that this is an interesting question that could be addressed in future studies. We have now briefly mentioned this aspect in the revised manuscript (**lines 526-527**). Finally, as per the reviewer's request, we have performed a ZFP30 and *Pparg2* reporter co-transfection (**Supplementary Fig. 3E**). We observed that ZFP30 tended to increase the activity of the *Pparg2*-P but not the *Pparg2*-PdM reporter (as now detailed in **lines 355-360**), consistent with other results (including novel CRISPR-based data) pointing to ZFP30 acting as an activator at this locus.

5. Figure 6C requires western blots to demonstrate that the level of KAP1 protein is similar across various mutants. These data are necessary to distinguish between effects mediated by 473 phosphorylation and KAP1 expression differences.

Response: We have now determined the expression of KAP1 by Western Blotting, as shown in the revised **Figure 6C**, as requested. No obvious differences were observed.

6. Forced expression of several ZFPs promoted 3T3-L1 adipogenesis in the Gubelmann paper. Do these ZFPs share a common mechanism of action with ZFP30? The authors should discuss whether the biology of ZFP30 is likely to be distinct within this group.

Response: Indeed, our screen revealed several ZFPs as regulators of adipogenesis, including the top three ones in terms of their regulatory effect. As part of our larger project, one of our initial aims was to discover potential common mechanisms of action among these three TFs (ZEB1, ZFP277 and ZFP30). However, we discovered that (1) ZEB1 has a very broad target landscape and largely co-localizes with other adipogenic TFs such as C/EBPbeta, while directly targeting both early and late known adipogenic regulators; (2) ZFP277-HA does not localize to the cell nucleus, thus its mechanism of action may not be transcriptional; (3) ZFP30 showed a very specific and restricted binding pattern, and the most coherent regulatory signal in the context of adipogenesis pointed to direct *Pparg* regulation. Thus, broadly speaking, there were no parallels that emerged between these factors. Interestingly, we note though that ZEB1 and C/EBPbeta also appear to bind to the ZFP30-bound enhancer that we have characterized in the current study, suggesting that this element may act as a regulatory hotspot. We have now incorporated this notion into the discussion section of the revised manuscript (**lines 528-546**).

Reviewer #3 (Remarks to the Author):

The manuscript by Chen et al provides molecular details of how a transcription factor (TF) called ZFP30 regulates adipogenic differentiation. ZFP30 belongs to a family of Krüppel-associated box Zinc Finger Proteins (KZFP), which are encoded by about 600 genes in the mouse genome. The canonical model suggests that KZFP bind specific DNA sequences within transposons and recruit KAP1 co-repressor to downregulate the target retrotransposon. The manuscript describes an atypical mode of action of ZFP30 to make two novel claims: (i) ZFP30 recruits KAP1 to activate rather than repress one target retrotransposon. (ii) Since this retrotransposon is located within an enhancer element of PPARG2 locus, ZFP30 activates PPARG2 gene, and in turn promotes adipogenic differentiation. The findings reported in this MS are interesting given the paucity of our understanding of the largest family of TF, especially in the regulatory context beyond retrotransposon silencing. However, both the claims made in the MS require further substantiation by performing some critical experiments. This is important since ZFP30 has already been identified as an adipogenic regulator by the authors' previous study (Gubelmann et al 2014). Thus the current MS is expected to identify detailed molecular mechanism by which ZFP30 regulates adipogenesis. The critical points that need authors' attention are summarised in the following:

1. The claim that ZFP30 activates retroelement/ gene expression via KAP1 is rather circumstantial. First, the downregulation of ZFP30 targets upon ZFP30 loss-of-function is much less frequent than upregulation (Line 258). It appears that the major role of ZFP30 is to repress targets, and not to activate them. By calling this phenomenon "context-specific" (line 238) the authors ignore the broad function of ZFP30 in repressing genes to focus on a single location at which ZFP30 may activate the gene (line 349). This case appears more like an indirect effect/ an exception than a canonical function of this ZFP30. In such a scenario, it is imperative to explain why ZFP30 behaves dramatically opposite depending on the target locus. E.g. is there a co-occurring TF? Are these indirect effects? It is also important to show the effect of ZFP30 knock-down on the expression of retroelements (by re-analyzing RNAseq) which are completely ignored despite being central to authors' narrative. By RNAseq it will be difficult to distinguish different instances of retroelements in the genome, but unique reads in histone marks ChIPseq / unique primers in ChIP-qPCR can directly assess changes in expression of ZFP30-targeted retroelement instances in undifferentiated and differentiated cells. In line with this, quantitation of retroelement-neighborhood effects on gene expression should be done to explain what proportion of ZFP30 peak neighbours are up- or down-regulated and distance between the promoter and ChIP peak. Is there a CTCF site in between the two that explains why some neighbors are upregulated and others are downregulated? Cell-line-specific differences noted by the authors further complicate the simple interpretation that ZFP30 activates the target genes. An explanation/ analysis of why certain targets are regulated cell-line-specifically is required.

Response: Genome-wide RNA-seq experiments have shown that TFs have both UP/DOWN effects on their target genes, irrespective of their traditional characterization as “repressors” and “activators”. For example, as discussed by Lambert et al. (Cell, 2018): “TFs have traditionally been classified as either “activators” or “repressors”; however, this notion has been repeatedly questioned. Many TFs can recruit multiple cofactors that have opposite effects (Frietze and Farnham, 2011, Rosenfeld et al., 2006, Schmitges et al., 2016), dependent on the local sequence context and availability of cofactors (Meijsing et al., 2009, Wong and Struhl, 2011). MAX, for example, functions as an inhibitor when binding to DNA as a heterodimer with MNT or MXD1 and as an activator when binding as a heterodimer with MYC (reviewed in Amati and Land [1994]). A recent study used a complex pool of >4 million sequences to survey the effect on gene expression of the relative positions of various TF-binding sites in diverse contexts, uncovering numerous motifs capable of both activation and repression in the same cell type (Ernst et al., 2016).” Thus, we believe that ZFP30 makes no exception,

and has context-specific regulatory effects. This is also visible through the differences in gene expression direction at the same ZFP30 target genes in two distinct adipogenic settings (two preadipocyte lines: 3T3-L1 and IBA). With this in mind, we tightened our analysis by focusing on consistent differences across these two systems. As shown in **Figure 3C**, there are only 12 ZFP30-bound genes that show such consistent expression changes. We further validated the influence of ZFP30 on the expression of eight of these genes in an exogenous setting through luciferase assays, finding that 80% of them are indeed altered. Two of them were consistently downregulated when *Zfp30* was decreased - the master adipogenic regulator *Pparg* and the adipokine & lipid metabolism-regulator *Angptl4*. Thus, even in this stringent context, *Pparg* appears not to be a single exception.

We tried to explore the co-occurring TFs/CTCF as suggested by the reviewer, in order to uncover possible regulatory mechanisms. As already detailed above (reviewer 2, point 2b), we found some adipogenesis-associated motifs in ZFP30-bound regions, including NF1, FOSL2, ATF3/4, and C/EBPbeta (**Supplementary Table 5**). However, the low number of regulated genes (3 genes upregulated and 9 downregulated in **Figure 3C**) provide not sufficient power to computationally determine the context-specific effect. We thus resorted to finding the ZFP30 partner biochemically. We found that KAP1 interacts with ZFP30 and promotes adipogenesis similar to ZFP30. Strikingly, among the 12 regulated genes shown in **Figure 3C**, *Pparg* is the only one bound by KAP1 at the same locus where ZFP30 was shown to bind. We demonstrate that KAP1 recruitment to this locus depends on ZFP30 (**Figure 5E and Supplementary 5G-H**). This genomic locus shows enhancer activity, and this activity is regulated by both ZFP30 and KAP1 (**Figure 3H, 3I and 5F**). To further strengthen our conclusion, we knocked out the ZFP30 binding motif in the *Pparg* enhancer as suggested by the reviewer (see also reviewer 1 (major comment #16)). As shown in **Figure 3J-K and Supplementary Fig. 3F-G**, 4 out of 5 clones with the *Zfp30* binding site deletion exhibit much less lipid accumulation compared to the wild-type clones. As expected, *Pparg2* and *AdipoQ* gene expression are largely reduced in the targeted cells (**Figure 3K and Supplementary Fig. 3G**). These collective data strongly support the notion in our opinion that the ZFP30/KAP1 complex targets the *Pparg* locus in a direct manner.

We also analyzed the expression of TEs based on our RNA-seq data as requested. Similar to what we observed for genes, we found both upregulated and downregulated TEs upon *Zfp30* KD (**Supplementary Fig. 3A**). However, there is no overlap between the differentially expressed TEs and ZFP30 binding. These findings suggest that ZFP30 does not function as a direct TE repressor, a proposed primary function of KZFPs. This is further supported by the short length of ZFP30-bound L1s, suggestive of their decay (**Supplemental Fig. 3B**). Consistently, the TE bound by ZFP30 in the *Pparg* locus is a heavily decayed L1MC5a. We reasoned that the transcript from this TE would not be detected by the polyA enriched RNA-seq, even it is expressed. This is why we have performed RT-qPCR of this locus using a random hexamer for reverse transcription. As shown in **Supplementary Fig. 3D**, we indeed detected the TE-derived transcript, whose expression is substantially reduced in *Zfp30* KO cells (**lines 341-348**). These data are consistent with our conclusion that ZFP30 modulates the activity of the TE-derived enhancer in the *Pparg* locus.

In summary, our new data together with the original data all support the idea that ZFP30 directly activates a TE-derived enhancer to promote *Pparg2* expression and as such positively regulates adipogenesis. We provide a new mechanism for how a KZFP can assume a regulatory role that is distinct from its presumed, canonical function, which is gene repression. Although there are several genes that appear to be regulated by ZFP30, *Pparg2* is the main target in an adipogenic context, not only because of the master regulator role of PPAR γ , but also because of newly generated data showing that deletion of the single *Zfp30* binding motif in the *Pparg2* enhancer mimics the *Zfp30* depletion phenotype (**Figure 3J-K**) while ectopic expression of *Pparg2* rescues the differentiation defect in *Zfp30* depleted cells (**Figure 5H**). There could be multiple mechanisms regarding how ZFP30

may regulate gene expression, as correctly pointed out by the reviewer, in that loss of ZFP30 function results in both upregulated and downregulated genes. Even the genes that exhibit the same trend might be regulated in a different manner. A full understanding of the regulatory function of this so far largely uncharacterized TF will however require further study, which we deem, is beyond the scope of the current manuscript.

2. The link between ZFP30 with PPARG2 regulation in the context of adipogenesis must be further strengthened. First, if ZFP30 regulates adipogenesis via PPARG2 as claimed by authors, ectopic expression of PPARG2 should rescue differentiation defect in ZFP30/ KAP1 KO cells. This is especially relevant as KAP1 KO causes cell death in one cell line (Line 314), suggesting PPARG2 misregulation as a narrow interpretation of loss-of-function of ZFP30/ KAP1. Second, an experiment that would demonstrate regulation of endogenous PPARG2 by the retroelement is to delete the minimum possible sequence spanning ZFP30 motif in the endogenous locus of PPARG2 enhancer. Most master regulators are themselves regulated in a complex manner by long-range interactions, which cannot be captured by enhancer-reporter constructs used by the authors. This experiment will also partially answer the criticism of indirect effect of ZFP30 loss-of-function.

Response: We have performed the rescue experiment suggested by the reviewer. We found that ectopic expression of *Pparg2* could efficiently rescue the differentiation defect both in *Zfp30* KO and *Kap1* KD cells (**Figure 5G-H**)(lines 460-462). These data corroborate the notion that *Pparg2* is the main target of ZFP30/KAP1 in this adipogenic system. We have also performed the ZFP30 binding motif KO in the *Pparg2* locus, as already detailed above (see reviewer 1 (major comment #16)).

3. The contribution of KAP1 to the regulation by ZFP30 adds to the complexity of the model. Since this is an important and non-classical finding for future research and KAP1 has already been linked with another adipogenic regulator C/EBPbeta (PMID 9742105), KAP1-ZFP30 nexus in adipogenesis requires further clarifications. First, the reduction in KAP1 binding to PPARG2 locus upon ZFP30 knock-down can be simply explained by lack of open chromatin at this locus without ZFP30, and not direct recruitment of KAP1 by ZFP30 as claimed by authors. To directly demonstrate that ZFP30 recruits KAP1 to PPARG2 locus the author should utilize specific domain deletions that uncouple a KZFP from binding to KAP1. Rescue of ZFP30 knock-out cells by such mutants will be useful in showing a direct recruitment of KAP1. Second, the phosphorylation on KAP1 seems to be unlinked to the proposed mechanism, and appears only circumstantial. An increase in global phosphorylation of KAP1 would have effects on all KAP1 targets and not just PPARG2. Does ZFP30 bind more effectively to phospho-KAP1? Does ZFP30 binds targets more efficiently in presence of phospho-KAP1? While identity of the kinase is important, it will be beyond this MS. Nonetheless, the importance of the phosphorylation in regulating PPARG2 may be indirect and independent of ZFP30. Efforts and experiments should be made to integrate phosphorylation data in the rest of the manuscript, else can be skipped altogether.

Response: To address this important point, we generated a ZFP30 mutant (In24aa) with an insertion of 24 additional amino acids in the KRAB domain derived from the allele 2 of *Zfp30* KO clone c15 (**Supplementary Fig. 1C**). As shown in **Figure 4C**, while the In24aa mutant contains the intact DNA binding zinc finger domains, it loses the capacity to interact with KAP1. We introduced this mutant as well as wild-type ZFP30 into the *Zfp30* KO IBA cells, respectively (**Supplemental Fig. 5G**). As shown in **Supplementary Fig. 5H**, re-expression of wild-type ZFP30 restored the recruitment of KAP1 to the *Pparg2* locus, while the ZFP30-In24aa mutant failed to do so. These data exclude the possibility that ZFP30 is responsible for opening the chromatin and in doing so, allowing KAP1 to be recruited by other factors. Rather, our findings support the idea that ZFP30 recruits KAP1 directly (**lines 449-453**).

We agree with the reviewer that the global phosphorylation of KAP1 would have effects on many KAP1 targets and not just *Pparg2*. We indeed tried to profile the global targets of S473 phosphorylated KAP1 by ChIP. However, neither the commercial antibody (BioLegend, 644602) nor our two batches of customized antibodies could successfully ChIP S473 phosphorylated KAP1. We

thus focused our efforts on *Pparg2*, which we already knew to be a main target of ZFP30/KAP1 (see our response to the first concern of reviewer #3). In our original manuscript, we showed that KAP1 is phosphorylated at S473 during adipogenesis (**Figure 6A-B**), and that this phosphorylation is involved in regulating adipogenesis (**Figure 6C**). Using luciferase assays, we showed that the phosphorylation mimicking mutant (S473E) activates *Pparg2* enhancer activity (Pparg2-P) (**Figure 6D**), while it fails to activate the reporter without the ZFP30 binding site (Pparg2-PdM) (**Figure 6D**). These data indicate that the function of KAP1 S473 phosphorylation in enhancing *Pparg2* expression depends on ZFP30. To further strengthen our conclusion, we examined the reporter activity in both *Zfp30* wild-type and KO cells. As shown in **Figure 6E** in the revised manuscript (**lines 493-496**), while KAP1 S473 activates the *Pparg2* reporter (Pparg2-P) in wild-type IBA cells, this capacity is impaired in *Zfp30* KO cells. Taken together, our data demonstrate that KAP1 S473 phosphorylation regulates *Pparg2* activation in a ZFP30-dependent manner. We agree with the reviewer that understanding how S473 phosphorylated KAP1 together with ZFP30 activates gene expression is of great interest. While we have excluded the possibility that S473 phosphorylation impacts the KAP1-ZFP30 interaction (**Supplementary Fig. 6** in the revised manuscript), addressing this important question is highly challenging at this stage, especially given that there is not an ideal S473 phosphorylation-specific antibody for IP and ChIP experiments. We therefore believe that the reviewer will agree with us that answering this question is beyond the scope of the current manuscript at this point.

4. Regulation and specificity of ZFP30: It is not clear in which tissues is ZFP30 expressed in comparison with adipogenic tissue. Also authors show that ZFP30 mRNA levels decrease upon adipogenic differentiation (Fig. S1D), not expected from a TF that directly regulates the differentiation. In the light of high levels of ZFP30 prior to differentiation, why should it activate PPARG2 gene only after the differentiation sets in, when in fact the levels of ZFP30 go down? If anything, ZFP30 may assist other TF such as C/EBPbeta/ delta in activating PPARG2. The DNase hypersensitivity data suggest that ZFP30 binds to many targets after differentiation as a consequence of chromatin opening by other pioneer TF than a primary cause of the differentiation process (Fig. 2I/ J). Hence the function of ZFP30 must be explained in the context of regulation by other known TF. After all, as the authors themselves claim that adipogenic regulation 'is one of the better characterized differentiation networks'. The study will be much better received if the function of ZFP30 is integrated in the known differentiation network of adipogenesis.

Response: We thank the reviewer for encouraging us to dig deeper into ZFP30's relation with other adipogenic TFs. *Zfp30* is expressed in multiple tissues at variable levels (**Supplementary Fig. 1I**). This, together with our RNA-seq data (**Figure 2E**), suggests that ZFP30 might have additional functions beyond regulating adipogenesis. While *Zfp30* is already highly expressed before differentiation, it binds and activates *Pparg2* only after differentiation. This suggests that ZFP30 is not a pioneer factor and that this TF is instead dependent on other (pioneer) factors to access the chromatin. This is consistent with the fact that ZFP30 preferably binds to DNase I hypersensitive loci, as pointed out by the reviewer and also shown in **Figure 2J**. Moreover, there are several adipogenic transcription factor binding motifs co-enriched in ZFP30 binding sites (**Supplementary Table 5**) and ZFP30 shares a large fraction of binding loci with these factors, which for example include the adipogenic pioneer factor C/EBPbeta and ZEB1 (Gubelmann et al., eLife, 2014) (**Supplementary Fig. 2R**). Based on these collective results, we propose that ZFP30 is a highly specific but integral part of the wider adipogenic gene regulatory network: its binding is facilitated by prior chromatin opening by C/EBPbeta or other factors, while it itself recruits KAP1 to increase *Pparg2* expression in an S473 phosphorylation-dependent manner. We have now better integrated these findings into the discussion section of the revised manuscript (**lines 528-545**).

Minor points:

1. The abstract states 'adipogenic exaptation'. This is overinterpretation of the data and not necessary. The 'switch' in function of ZFP30 from repressor to activator has no basis. Such ideas should be reduced to a couple of lines in discussion.

Response: We have modified the abstract as suggested, leaving this notion now only to the discussion where it can be better conceptualized (**lines 24-26 and 572-575**).

2. Line 107: What is MDI cocktail? No full form is given in the main text.

Response: We have added the full form in the text of the revised manuscript (**lines 112-113**).

3. Fig. 3B: The color code of Day 0 and Day 2 in control and KO/ KD is confusing. Also Fig. 3D: ratio of control to shRNA is non-intuitive, as against the normal shRNA to control ratio (where upregulation is positive and downregulation is negative).

Response: We have now modified the figures as suggested.

4. Line 293: "Note that we define..." does not make sense in the context of the interpretations just before this line.

Response: We have now reworded the text to avoid the stated confusion in the revised manuscript (**lines 353-360**).

5. Line 306: 3T3-L1 cell line is not a 'physiological condition'.

Response: We have changed to "adipogenic environment" in the revised manuscript.

6. Line 337: How do the 500 peaks of KAP1 compare with other cell types such as ESC? Thousands of peaks were reported in previous studies.

Response: The KAP1 expression level varies in different cells. It is especially highly expressed in ESCs (see below the expression level from BioGPS). This potentially explains the greater number of peaks detected in these cells to some extent. We note however that a correlation analysis between the CHIP enrichments of *Kap1* as well as IgG control replicates (**Supplementary Fig. 5B** in the revised manuscript) showed a clear separation of CHIP-ed factor first, and days second, supporting the quality of our data.

Figure legend: Expression of *Kap1* in different mouse cells/tissues from BioGPS.

7. Line 401: Typo error ZPP30.

Response: Thank you for spotting it. We have now corrected it in the revised manuscript.

8. Line 472: How are *Nnat*, *Plagl1* and *Peg3* regulated under the conditions of Loss of function of ZFP30?

Response: We have verified the expression of these three genes. *Peg3* is the only one that is differentially expressed in *Zfp30* KD cells. This is now incorporated in the discussion of the revised manuscript (lines 600-601).

REVIEWERS' COMMENTS:

Reviewer #1 (Remarks to the Author):

I congratulate the authors on the significant improvement of the manuscript. The message is better conveyed now and reinforced by the new experiments and analyses.

Reviewer #2 (Remarks to the Author):

The authors have performed several technically challenging experiments that improve the study by addressing most of my concerns and those of the other reviewers. Congratulations on a job well done.

Reviewer #3 (Remarks to the Author):

The authors have done an extensive revision, answering almost all the questions that were raised. I would only like to suggest to authors to re-consider their decision on including the phosphorylation data in the current MS. The data seem still very loosely attached to the rest of the narrative, and do not add much to the model. Instead the authors should remove the phospho data, and send it out in another MS after working on the phosphorylation link in better detail. My major concern is that the superficial phospho data currently reported in the MS will reduce the clarity of the otherwise solid paper, raising further questions.

REVIEWERS' COMMENTS:

Reviewer #1 (Remarks to the Author):

I congratulate the authors on the significant improvement of the manuscript. The message is better conveyed now and reinforced by the new experiments and analyses.

Reviewer #2 (Remarks to the Author):

The authors have performed several technically challenging experiments that improve the study by addressing most of my concerns and those of the other reviewers. Congratulations on a job well done.

Response: We thank the reviewers for their support and efforts to review our manuscript.

Reviewer #3 (Remarks to the Author):

The authors have done an extensive revision, answering almost all the questions that were raised. I would only like to suggest to authors to re-consider their decision on including the phosphorylation data in the current MS. The data seem still very loosely attached to the rest of the narrative, and do not add much to the model. Instead the authors should remove the phospho data, and send it out in another MS after working on the phosphorylation link in better detail. My major concern is that the superficial phospho data currently reported in the MS will reduce the clarity of the otherwise solid paper, raising further questions.

Response: We thank the reviewer for her/his support and constructive suggestions. As per the reviewer's and editor's request, we have removed the KAP1 phosphorylation data in the re-revised manuscript.